# A Novel Single Differencing Measurement for Multipath Detection

Matthew Alcock *,† and Paul Blunt †

Nottingham Geospatial Institute, The University of Nottingham, Nottingham NG7 2TU, UK; paul.blunt@nottingham.ac.uk
* Correspondence: matthew.alcock@nottingham.ac.uk
† These authors contributed equally to this work.

**Abstract:** Increased global dependence on Global Navigation Satellite Systems (GNSSs) has resulted in a high demand for greater precision and reliable measurements from GNSS receivers. The multipath problem is the single largest source of errors in modernised GNSSs. Double differencing techniques, such as Code Minus Carrier (CMC) have been shown to accurately detect and measure multipath, allowing for corrections to be made via Ground Base Augmentation Systems (GBAS), for example. However, these techniques require at least two receivers and the protection provided is not extended to stand-alone receivers. This paper introduces a new single differencing technique for the accurate detection of multipath in standalone GNSS receivers receiving modernised Binary Offset Carrier (BOC)-modulated signals. Similarities to CMC are drawn before the novel measurement, Code minus Subcarrier, (CMS) is characterised statistically and a threshold for multipath detection is determined. The effectiveness and sensitivity of this novel measurement as a multipath detection technique are analysed through simulation and multipath error envelope analysis. It will be shown that multipath echos capable of inducing a psuedorange error larger than the threshold are detectable at any amplitude. The method is finally verified using simulated fixed offset multipath, confirming that when code and subcarrier early–late spacings are optimal, all ranges of multipath delays, even as small as 21 meters, are detectable. This novel method of multipath detection requires no additional complex correlators than already exist in the chosen tracking algorithm, thus, providing excellent detection with minimum complexity added to the receiver structure.

**Keywords:** multipath; detection; differencing; BOC; modernised GNSS

## 1. Introduction

Global Navigation Satellite Systems (GNSSs) such as the American Global Positioning Service (GPS), European Galileo and the Chinese Beidou system are satellite based systems that provide accurate position and time information, traditionally to users on Earth. Initially GPS was intended for military purposes, however it has been heavily adopted by a wide array of civilian applications, such as maritime, surveying, agriculture, aviation and finance, to the point where there is now a global dependence on GNSS. This dependence combined with GNSS inherent vulnerabilities are of great concern to governments and business leaders alike. For instance, in 2000 The GNSS Safety and Sovereignty Convention of 2000 AD was created to translate these technical vulnerabilities into an admissible multilateral treaty [1]. Many of these applications require a reliable Position, Navigation and Timing (PNT) solution. This is especially true when services are concerned with Safety of Life (SoL). The reliability/performance of GNSS can be deteriorated by various factors including, ephemeris errors, atmospheric errors and radio frequency (RF) interference. RF interference can be classed as unintentional (out-of-band emission), e.g., WiFi and 4G or intentional (in-bound), e.g., jamming and spoofing [2]. Receivers themselves introduce error to the GNSS measurements through the inherent system noise, determined by the

noise figure of said system, and clock biases as a result of using cheap/small crystalline clocks. Many of these errors can be reduced or completely mitigated through modelling, multi-frequency and/or single/double differencing techniques [2]. One major error source that is difficult to model and reliably measure is multipath. This is such that multipath is considered to be the dominant restricting factor in precision-oriented GNSS applications [3]. The detection of multipath in stand-alone receivers is the focus of this work.

Multipath occurs when the transmitted, Line-of-Sight (LoS) signal is reflected off nearby obstacles, such as buildings or metal surfaces near the receiver. These reflections can be either specular (reflected from a smooth surface) or diffuse (reflected off a rough surface) and arrive at the receiver attenuated and delayed with respect to the LoS signal, as the reflection increases the path length and alters the carrier phase/frequency and polarisation [4]. When superimposed on the LoS signal, the signal becomes distorted and a bias is introduced into the signal time delay estimate output from the Delayed Lock Loop (DLL), within the tracking unit of the receiver. This leads to a bias in the psuedorange and errors in the resulting PNT solution, reducing the precision of the formed solution. The LOS signal component can be blocked completely, causing the receiver to form position solutions using the Non-Line-of-Sight components; however, the focus of this work is on the combination of multipath echoes and LoS signal at the receiver antenna. The combination of these signals within the receiver cause the code correlation function in the DLL to become distorted, which in turn shifts the discriminator zero crossing and tracking point, causing large ranging errors. This can be seen in Figures 1 and 2, where a single multipath echo with a delay of 0.25 chips and a relative (to LoS) amplitude of 0.5 is superimposed onto the LoS signal and resulting correlation/discriminator curves are formed accordingly.

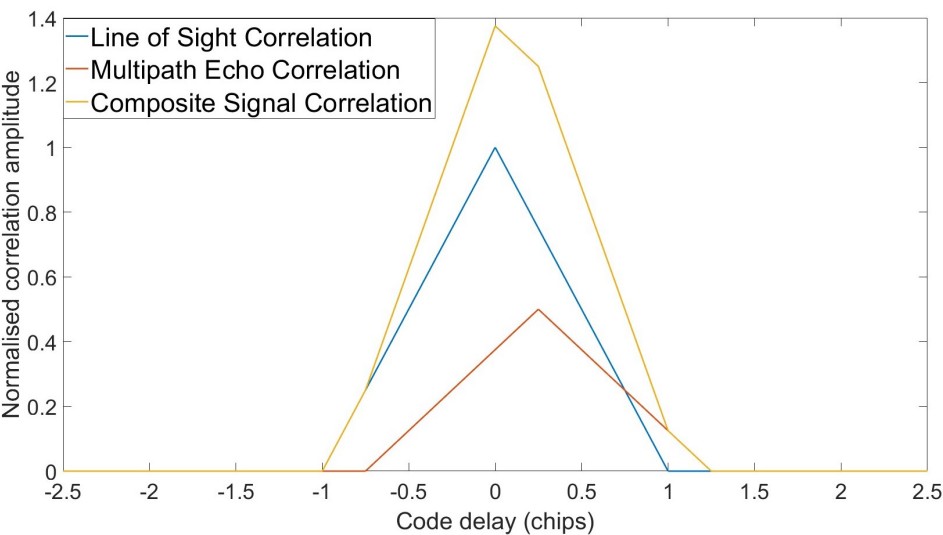

**Figure 1.** Direct, multipath and composite signal correlation functions.

The multipath problem has fascinated researchers from numerous communication-based fields for many years, and as a result, multiple methods of detection and mitigation have been developed. A thorough discussion of a large selection of these techniques can be found in [3]. GNSS Signals offer a form of inherent protection against multipath, in that they are Right-Hand Circularly Polarised (RHCP) and after a reflection from a smooth surface (specular) the polarisation is reversed. Thus, a receiver using an RHCP antenna would heavily attenuate these single reflections; however, signals that reflect off a rough surface (diffuse) are randomly polarised and are only marginally attenuated by an RHCP antenna [5]. There is also the fact that most mobile devices use linearly polarised antennas that offer no such protection. Choke rings can mitigate multipath as their grooves cancel out the strongest elements of the reflected signal as shown in [6]. Ref. [7] presented the Multipath Limiting Antenna (MLS), which attenuates low-elevation-angle signals (ground reflections will come to the receiver from low elevation) through use of a high-zenith

antenna (HZA). Phased arrays and beam forming have also been shown to be effective at mitigating multipath [8].

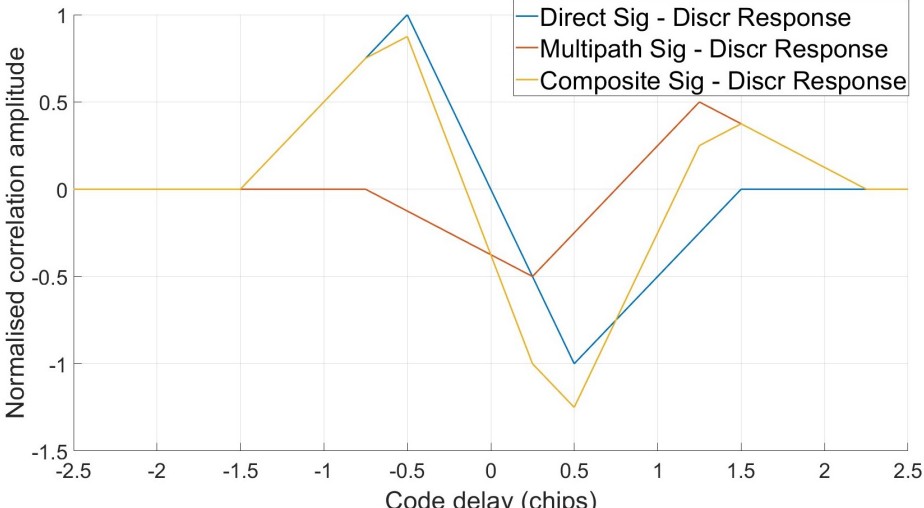

**Figure 2.** Direct, multipath and composite signal discriminator functions.

Antenna techniques have proved effective; however, their use is limited to those systems that can afford the additional hardware and costs associated with them. Receiver-based technologies offer detection and mitigation without the need for additional hardware and seem more suitable for stand-alone receivers. Receiver multipath detection/mitigation techniques typically monitor the code correlation function within the DLL or the raw measurements from the code loops. One of the most common feedback delay estimators is the Early-Minus-Late (EML) DLL. A single pair of correlators, spaced one code chip apart, are used to form a discriminator function, whose zero crossings determine the path delays of the received signal. As seen in Figures 1 and 2, the presence of multipath distorts the correlation and discriminator functions. In [9], narrowing the spacing between the correlators was shown to dramatically reduce the ranging errors caused by multipath. However, the choice of correlator spacing is largely determined by the available front-end bandwidth [2] and the environment the receiver is to be operating in (highly dynamic environments require a wider spacing). Double Delta Correlator is a term that encompasses the various special discriminators which are formed by two or more pairs of correlators, instead of one. The process is described in [10], which introduces the High-Resolution Correlator. Various Signal Quality Monitoring (SQM) techniques, including Double Delta have been investigated for multipath mitigation in [4] and Franco-Patiño et al. [11]. These multi-correlator techniques add computational complexity and their implementation tends to be associated with specific receiver signal operation, for example Ashtech's Strobe CorrelatorTM and NovAtel's Pulse Aperture CorrelatorTM (PAC). At a raw measurement level, a single differencing technique known as Code Minus Carrier (CMC), has been shown to accurately measure code multipath in [12] and is used in Ground-Based Augmentation Systems (GBAS) all over the world [13]. CMC, which will be explored in more detail later, however, requires multiple receivers and suffers from the fact that as well as the code multipath the CMC measurement also contains twice the ionospheric delay. However, using dual-frequency receivers and time differencing the measurements it was shown in [14] that it is possible remove this bias and use the measurements for multipath detection and measurement.

This paper introduces a novel single differncing measurement that takes advantage of the modernised GNSS signal modulation, Binary Offset Carrier BOC(fs, fc). BOC modulation adds a fourth component, a deterministic square wave subcarrier, to the three existing components of Binary Phase Sift Keyed (BPSK) signals such as GPS L1 C/A. BPSK modulates the data signal onto a Pseudo Random Noise (PRN) code sequence that is

modulated onto the carrier at the desired centre frequency. The separate components of BOC modulation (ignoring the data) are shown, not to scale in Figure 3. This novel measurement will subtract the subcarrier phase from the code phase measurement. The subcarrier loop has been shown to reduce the noise on the code loop through a Subcarrier-Aided Code Tracking (SACT) scheme using the DE tracking method in [15]. Ref. [16] uses a BPSK-like tracking algorithm to achieve similar code smoothing. In [17] it was shown that dynamics of the code and subcarrier divergence are very low; thus, subcarrier aiding can be used with a very narrow code loop bandwidth.

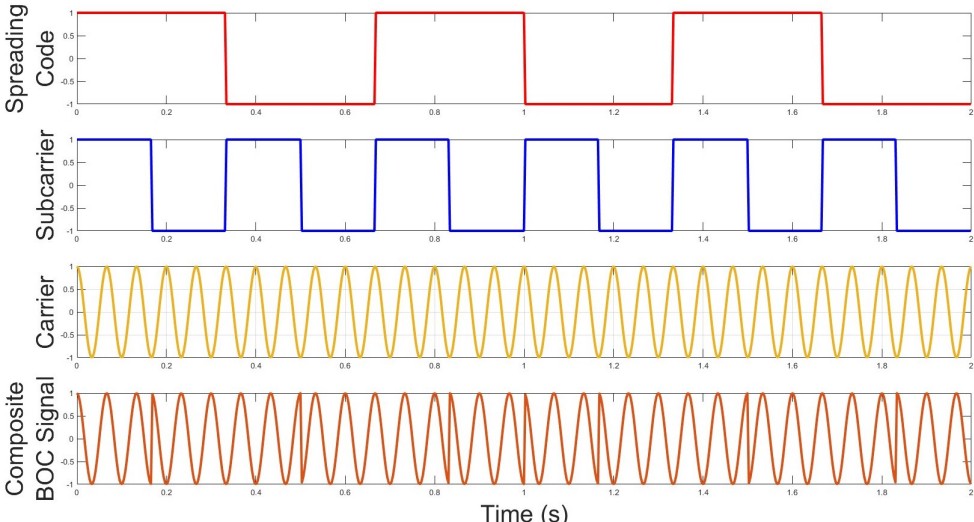

**Figure 3.** BOC signal components.

The subcarrier is modulated onto the code, and its frequency, fs, is an integer multiple of 1.023 MHz (the code base chipping rate) and the code chipping rate, fc, is also a multiple of of 1.023 MHz [18]. BOC modulation has been used for all Galileo Open Service Signals and is being used in the modernisation of GPS signals and Beidou B1 (L1) signals. In this paper we will briefly revisit BOC modulation and a signal model will be presented. As it is the chosen modulation scheme for most modernised GNSS signals, it seems appropriate to develop measurements/techniques that make best use of its full potential. This will lead to a brief discussion on a range of suitable BOC tracking schemes before selecting one to work with for the rest of the paper. The following section will introduce and form the Code Minus Subcarrier (CMS) measurement. The measurement will be characterised statistically using simulated data and its effectiveness and sensitivity as a multipath detection method will be analysed through simulation and multipath error envelope analysis. Utilisation of the measurement for multipath detection will be verified using fixed-offset multipath using simulation data.

## 2. Materials and Methods

The organisation of this paper, discussed above, is illustrated in Figure 4. This section will focus on the initial three tasks and the remainder will be addressed in the results and and conclusion sections.

### 2.1. BOC Signal and Unambiguous Tracking

As discussed in Section 1 BOC modulation was selected for modern GNSS signals including Galileo and modernised GPS. BOC was chosen for Galileo as modulating the subcarrier onto the existing BPSK signal has the effect of splitting the main beam of the Power Spectral Density (PSD) into two main lobes, equally spaced about the BPSK centre frequency. This allows for the transmission and reception of modernised signals on the same centre frequency of existing legacy signals [19]. This can be seen in Figure 5,

which shows normalised PSDs for BPSK(1)—representative of the GPS C/A signal, BOC(1, 1)—representative of the major component in the E1B signal and BOC(2, 1) having the same subcarrier to code ratio as the E6 Public Regulated Service (PRS) signal, which uses BOC(10, 5) modulation.

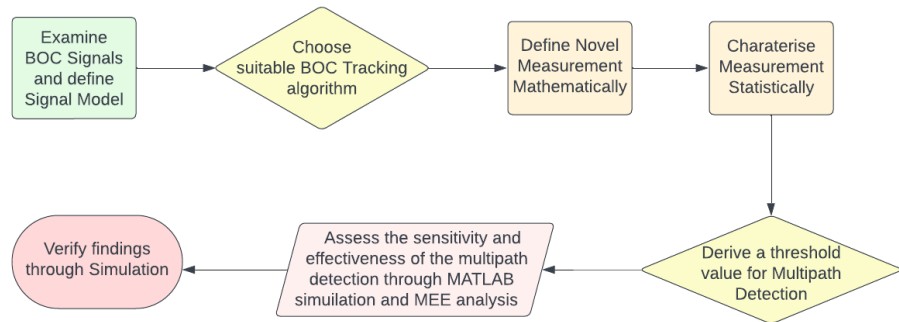

**Figure 4.** Organisation of this paper.

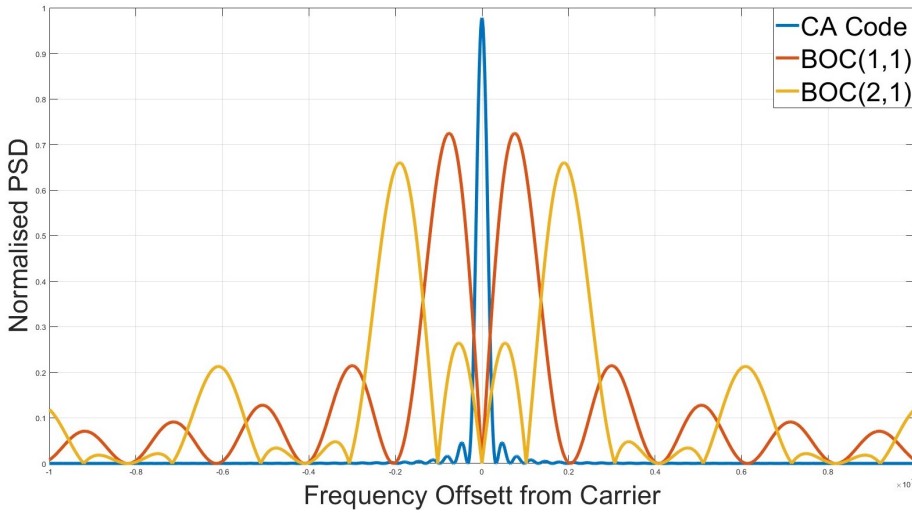

**Figure 5.** Normalised power spectral densities of BPSK(1), BOC(1, 1) and BOC(2, 1).

As well as the improved spectral sharing, BOC has many other advantages over BPSK, including reduced mutual interference on common carrier frequencies and improved signal tracking accuracy due to the high frequency content. This improved tracking accuracy, though, does come with a trade-off in that the tracking becomes ambiguous due to the subcarrier's effect on the correlation function. Figure 6 shows normalised correlation functions for BPSK(1), BOC(1, 1) and BOC(2, 1). It can be seen that BOC signals have additional peaks in the correlation function when compared to the single-peaked BPSK triangle function. It is also evident that increasing the fs frequency term relative to the fc term increases the amount of peaks and thus ambiguity, unless carefully designed GNSS receivers tracking BOC signals could easily lock onto one of these false peaks, resulting in large pseudorange errors. These additional peaks mean the tracking of the composite BOC signal becomes ambiguous.

Many tracking algorithms have been designed to track BOC signals and combat the ambiguous nature of their correlation functions. These algorithms can be broadly categorised into techniques that treat the local code and subcarrier as combined (i.e., 1D correlation) and driven by the same Numerically Controlled Oscillator (NCO) and those that treat the code and subcarrier as separate (2D correlation) tracking the subcarrier independently using an additional control loop. Techniques that fall under the first category include Bump Jumping, BPSK-Like and Auto-correlation Side-Peak Cancellation Technique

(ASPECT). The Bump Jumping algorithm, presented in [20] uses two additional correlators (Very Early and Very Late) situated at the predicted locations of the side peaks, ensuring the prompt remains on the highest peak. BPSK-like algorithms see the BOC signals as the sum of two BPSK signals [21] and ASPECT removes side peaks of the BOC auto-correlation function to remove ambiguity [22]. However these algorithms do not make the most of the high-accuracy provided by the high frequency subcarrier. Algorithms such as the Double Estimator (DE) first presented in [23] treat and track the subcarrier independently such that in addition to the code and carrier control loops the DE adds an additional subcarrier loop. As the subcarrier is a square wave it can be tracked using a Delayed Lock Loop (DLL) much the same as the code, with Early, Prompt and Late correlations forming a discriminator function. The additional tracking loop is often referred to as the Subcarrier Locked Loop (SLL). The final code delay estimate, $\tau$, given by the DE is calculated using Equation (1).

$$\tau = \tau_{sc} + T_{sc} \cdot round(\frac{\tau_c - \tau_{sc}}{T_{sc}}) \tag{1}$$

where $\tau_{sc}$ is the accurate but ambiguous SLL delay estimate, $\tau_c$ is the less accurate but unambiguous DLL delay estimate and $T_{sc} = \frac{1}{2f_{sc}}$. The rounding operation assures that if the subcarrier tracking is not locked to the correct peak, the estimate is not affected as long as the code tracking error is smaller than half a subcarrier chip. The final estimate is a least mean square error solution and fully exploits the subcarrier accuracy provided by the BOC modulation [23]. A variation of the DE algorithm is the Double Phase Estimator (DPE), presented in [24]. The DPE replpaces the DLL used in the subcarrier with a Phase-Locked Loop (PLL). The DE and DPE both provide satisfactory unambiguous tracking and so the DE algorithm will be used for the analysis of the proposed measurement.

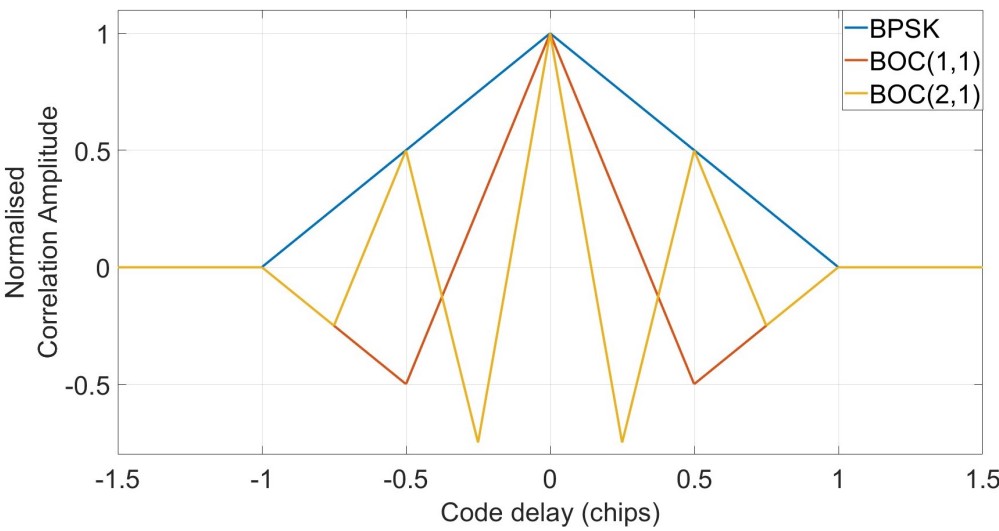

**Figure 6.** Normalised correlation functions of BPSK(1), BOC(1, 1) and BOC(2, 1).

### 2.2. Signal Model

This papers focus is on BOC modulated signals, specifically making use of the additional tracking measurement they provide for the purpose of multipath detection. Ignoring navigation data bit transitions and RF front end bandwidth, these signals can be expressed as Equation (2). BOC signals undergoing multipath can be expressed as Equation (3).

$$S = \sqrt{P}C(\tau - \tau_0)B(\tau - \tau_0)\cos(2\pi f_0 t + \phi_0) + \eta(t) \tag{2}$$

$$S = \sqrt{P}C(\tau - \tau_0)B(\tau - \tau_0)\cos(2\pi f_0 t + \phi) +$$
$$\sum_{m=1}^{n} \alpha\sqrt{P}C(\tau - \tau_m)B(\tau - \tau_m)\cos(2\pi f_m t + \phi_m) + \eta(t) \tag{3}$$

where

- $P$ = total received signal power;
- $C$, $B$ = PRN code and subcarrier signals, respectfully;
- $f$ = carrier frequency;
- $\phi$ = carrier phase;
- $\eta$ = thermal noise term (assumed to be zero mean Gaussian);
- $\tau$ = time delay;
- Subscripts $_0$ and $_m$ refer to line-of-sight and multipath signals, respectfully;
- Superscript $^n$ = number of multipath reflections;
- $\alpha$ = amplitude scaling factor.

The amplitude scaling factor, $\alpha$, can take values between 0 and 1, where 0 is no multipath and 1 is a multipath echo of equal amplitude compared to the LOS component.

To assess the effectiveness and sensitivity of the CMS measurement, Multipath Error Envelope (MEE) analysis will be performed. This will be discussed in more detail later however, to perform such analysis a BOC signal corrupted by a single multipath reflection of half the LoS amplitude is to be generated. This was performed in MATLAB using Equation (4), where the data modulation is ignored.

$$
\begin{aligned}
S = \sqrt{P}C(\tau - \tau_0)B(\tau - \tau_0)\cos(2\pi f_0 t + \phi) + \eta(t) \\
+\alpha\sqrt{P}C(\tau - \tau_m)B(\tau - \tau_m)\cos(2\pi f_m t + \phi_m) + \eta(t)
\end{aligned}
\tag{4}
$$

### 2.3. Code Minus Subcarrier Measurement

Differencing the subcarrier and the code measurements is not a completely new idea. It is essentially a evolution of an existing differencing technique, Code Minus Carrier (CMC), which uses an accurate and ambiguous measurement (the carrier phase) to smooth a noisy but unambiguous measurement, the code phase. Equations (5) and (6) show the measurement equations for the code and carrier phase, respectfully.

$$
\rho_{i,k}^s = R_k^s + c(\delta tu_k - \delta t_k^s) + I_{i,k}^s + T_k^s + MP_{i,k}^s + \epsilon_{i,k}^s
\tag{5}
$$

$$
\phi_{i,k}^s = R_k^s + c(\delta tu_k - \delta t_k^s) - I_{i,k}^s + T_k^s + N_{i,k}^s \lambda_i + mp_{i,k}^s + \zeta_{i,k}^s
\tag{6}
$$

where

- $R_k^s$ = geometrical distance from satellite 's' to receiver 'k', in meters;
- $c$ = free space velocity of light;
- $\delta tu_k$ and $\delta t_k^s$ = receiver and satellite clock biases, respectfully;
- $I_{i,k}^s$ = ionospheric delay in meters (corresponding to wavelength of frequency, $i$);
- $T_k^s$ = tropospheric delay in meters;
- $N_{i,k}^s \lambda_i$ = integer ambiguity (in cycles) and carrier wavelength of frequency $i$;
- $MP_{i,k}^s$ and $mp_{i,k}^s$ = code and carrier multipath, respectfully;
- $\epsilon_{i,k}^s$ and $\zeta_{i,k}^s$ = thermal noise errors on code and carrier measurements, respectfully.

Ref. [12] showed that differncing code and carrier measurements could isolate the code multipath component at the expense of doubling the ionospheric delay component and an additional carrier integer ambiguity term as seen in Equations (7) and (8).

$$
\text{CMC} = \rho_{i,k}^s - \phi_{i,k}^s = 2I_{i,k}^s + N_{i,k}^s \lambda_i + MP_{i,k}^s + \epsilon_{i,k}^s + mp_{i,k}^s + \zeta_{i,k}^s
\tag{7}
$$

The noise and multipath errors on the carrier are orders of magnitude smaller than that of the code and so the CMC measurement can be represented as Equation (8)

$$
\text{CMC} = 2I_{i,k}^s + N_{i,k}^s \lambda_i + MP_{i,k}^s + \epsilon_{i,k}^s
\tag{8}
$$

It can be seen that the common terms between the two measurements are cancelled resulting in a measurement containing the code multipath delay, twice the ionospheric

delay and the integer ambiguity term. The ionospheric delay term is doubled in a CMC measurement as it is frequency-dependent such that the code is delayed and the carrier is advanced in equal amounts [2]. These additional components can be estimated using dual-frequency techniques and removed by subtracting the mean from the observations. This works as the carrier phase ambiguity term remains constant during continuous tracking. This would not satisfy a stand-alone user in a city, where it was shown in [14] that time-differencing the CMC measurement can provide a multipath detection metric that does not require consistent tracking and thus useful in urban scenarios where blocking of signals is prevalent.

The Code Minus Subcarrier (CMS) measurement is much the same as CMC, except rather than differencing code and carrier phase measurements, both of the measurements are of code phase (i.e code and subcarrier). In [17], it was shown that subcarrier aiding can be used with a very narrow code loop bandwidth. Another key point to note from [17] is that the ionosphere is not a significant source of code/subcarrier divergence. This will be examined in Section 3.

The CMS measurement equations will now be presented in the same manner as CMC above, with Equations (9) and (10) representing the code and subcarrier measurements, respectfully.

$$\rho_{i,k}^s = R_k^s + c(\delta tu_k - \delta t_k^s) + I_{i,k}^s + T_k^s + MP_{i,k}^s + \epsilon_{i,k}^s \tag{9}$$

$$\varrho_{i,k}^s = R_k^s + c(\delta tu_k - \delta t_k^s) + I_{i,k}^s + T_k^s + mp_{i,k}^s + v_{i,k}^s \tag{10}$$

$$\text{CMS} = \rho_{i,k}^s - \varrho_{i,k}^s = MP_{i,k}^s + \epsilon_{i,k}^s + mp_{i,k}^s + v_{i,k}^s \tag{11}$$

This is similar to CMC, in that the code multipath has been isolated; however, noise on the SLL code loop is not an order of magnitude smaller like the phase noise is in relation to the code noise in the CMC measurement. The SLL noise is only $\frac{1}{2}\sqrt{fc/fs}$ of the DLL noise, assuming all parameters are equal [25]. Thus, methods of reducing the subcarrier noise need to be utilised to maximise the effectiveness of this measurement. To achieve this, the measurement must be characterised statistically, which is done in Section 3. By characterising the distribution of the CMC measurement under no-multipath conditions, where the measurement contains only noise from both sources, a threshold for multipath detection can be derived.

### 3. Results

#### 3.1. Code Minus Subcarrier Distribution

From Equation (11), it can be seen that under no-multipath conditions the CMS term is defined by the noise of the subcarrier and code loops. Both noise sources are zero mean Gaussian distributed, and the standard deviation of each is determined by the discriminator used in each loop. In this work, a non-coherent early minus late power discriminator is used for both code and subcarrier loops, the standard deviation of the noise in the code loop is given by Equation (12) and the deviation of the subcarrier by Equations (13) and (14) [2].

$$\sigma_{t_{DLL}} = \sqrt{\frac{BL_{co} \cdot d}{2c/n_0} \left[1 + \frac{2}{(2-d)(c/n_0)\tau_a}\right]} \tag{12}$$

$$\sigma_{t_{SLL}} = \frac{1}{2}\sqrt{fc/fs}\sqrt{\frac{BL_{co} \cdot d}{2c/n_0} \left[1 + \frac{2}{(2-d)(c/n_0)\tau_a}\right]} \tag{13}$$

For BOC(1,1) signals the noise on the subcarrier has a standard deviation as Equation (14).

$$\sigma_{t_{SLL}} = \sqrt{\frac{BL_{co} \cdot d}{8c/n_0} \left[1 + \frac{2}{(2-d)(c/n_0)\tau_a}\right]} \tag{14}$$

where $BL_{co}$ is the code loop bandwidth, $c/n_0$ is the carrier to noise density, $d$ is the early–late correlator spacing and $\tau_a$ is the accumulation time, which for Galileo E1B is typically 4 ms due to the 250 Bps data rate.

Thus, under no multipath the variance in the CMS measurement can be expressed as a combination of the variance in the two respective loops, using Equation (15) [26].

$$\sigma^2_{t_{CMS}} = \sigma^2_{t_{DLL}} + \sigma^2_{t_{SLL}} - 2cov(DLL, SLL) \qquad (15)$$

The loops cannot be said to be completely independent, as the 2D correlation surface was shown to be slightly skewed in [22], and thus the covariance between the two loops would need to be calculated for very precise applications, such as measuring the multipath on the code loop. The skew demonstrated in [22] was slight and thus the covariance will be small in relation to the variances. Thus, for the case of multipath detection, the metric variance can be assumed to be the sum of the two loop variances and the measurement standard deviation can be approximated as Equation (16)

$$\sigma_{t_{CMS}} = \sqrt{\sigma^2_{t_{DLL}} + \sigma^2_{t_{SLL}}} \qquad (16)$$

It can be seen that this approximation is an overestimate and its use as the basis of a multipath detection metric threshold will decrease the measurements sensitivity slightly. Therefore where precision and sensitivity are of high importance, measurement through a calibration campaign is essential. For the case of multipath detection in stand-alone receivers, though, this level of precision and sensitivity is unnecessary and the approximation in Equation (16) is satisfactory.

This has been verified through simulation. The simulator used is the Orolia GSG-8 and is capable of generating models that mimic the operation of multiple GNSS systems/constellations. In this case, a Galileo constellation, consisting of six visible satellites transmitting the E1B (MBOC) signal, is simulated in digital noise, with no atmospheric or ephemeris errors. The receiver antenna is modelled as an isotropic antenna so that elevation angles will not effect C/No, thus maintaining steady power levels throughout the run. Scenario parameters can be seen in Table 1. The Orolia GSG-8 was chosen as it is extremely versatile and generates all scenarios using software defined radios rather than the conventional Field-Programmable Gate Arrays (FPGA) techniques. Because of this, the GSG-8 can output the data as RF signals that can be processed using a front-end analogue to digital converter, or directly at digital base-band. The latter allows for experiments like this to perform tests without any of the front-end band limiting effects and has been selected for this specific experiment. The effect of the front-end filter will need to be, and is analysed in Section 3.

The no-error scenario was then processed through an amended version of the Soft-GNSS Software-Defined Receiver (SDR) designed in [27]. The SDR has been amended so it tracks signals in DE mode, initially tracking the carrier waveform in Frequency-Locked Loop (FLL) configuration before transitioning to a Phase-Locked Loop. Receiver parameters can be seen in Table 2. The code and subcarrier measurements are taken every 10th of a second and each satellite is tracked for 280 s. Thus, 2800 samples per satellite have been used to characterise the distribution of the CMS measurement, making 16,800 samples total. Within MATLAB, the function histfit was used to fit a normal distribution to the resulting histograms and normplot was used to assess the goodness of fit of those normal distributions by plotting both empirical and analytical Cumulative Distribution Functions (CDF). If the blue line which represents the empirical CDF fits to the red line which represents the analytical CDF, the distribution can be approximated by a normal distribution with measured mean and variance. The resulting distributions from two of the satellites can be seen in Figures 7 and 8.

**Table 1.** Simulation parameters for measurement distribution experiment.

| Simulation Parameter | Set Value |
|---|---|
| Sampling Frequency | 25 MHz |
| Atmospheric Errors | Off |
| Ephemeris Errors | Off |
| Constellation | Galileo E1B |
| Visible SVs | 2, 8, 11, 12, 24, 25 |
| Digital Noise | On |

**Table 2.** SDR parameters for measurement distribution experiments.

| Receiver Parameter | Set Value |
|---|---|
| Sampling Frequency | 25 MHz |
| FLL Noise Bandwidth | 2 Hz |
| PLL Noise Bandwidth | 10 Hz |
| DLL Noise Bandwidth | 0.5 Hz |
| SLL Noise Bandwidth | 0.5 Hz |
| FLL Discriminator | Cross over Dot Product |
| PLL Discriminator | 2 Quadrant Arc-tangent |
| DLL Discriminator | Early Minus Late Power |
| SLL Discriminator | Early Minus Late Power |
| DLL Early-Late Correlator Spacing | 0.5 Chips |
| SLL Early-Late Correlator Spacing | 0.25 Chips |
| Tracking Correlation Time | 4 ms |

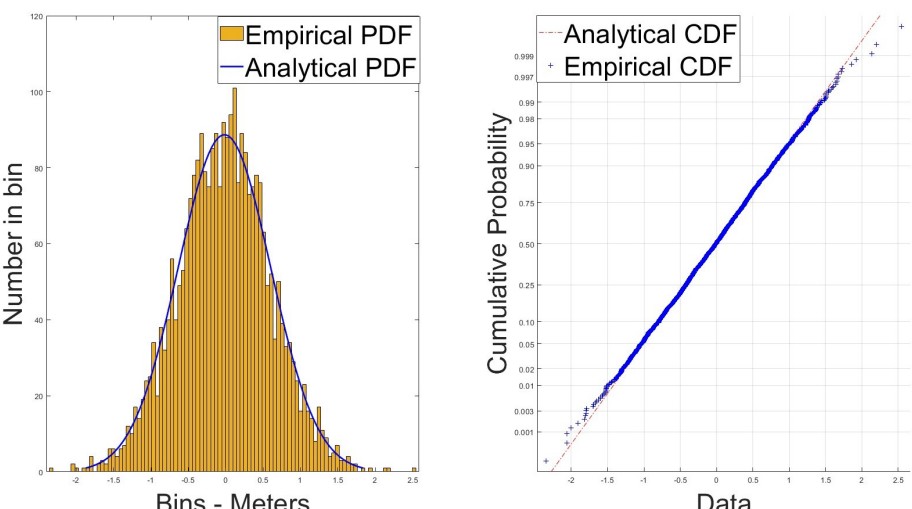

**Figure 7.** CMS measurement distribution—PRN 8.

Figures 7 and 8 clearly show that the CMS measurement can be characterised by a normal distribution with a mean and variance equal to that of the measured data. This was additionally examined using the lillietest function within MATLAB. Lillietest returns a test decision for the null hypothesis that the data in vector x comes from a distribution in the normal family, against the alternative that it does not come from such a distribution, using a Lilliefors test. The result is 1 if the test rejects the null hypothesis at the 5% significance level, and 0 otherwise. The result was zero for each PRN examined, indicating again that CMS measurement can be approximated with a normal distribution. Something of interest, though, was that the means of the analysed data was not zero (as expected). Table 3 shows the expected and calculated means and standard deviations for each PRN under test.

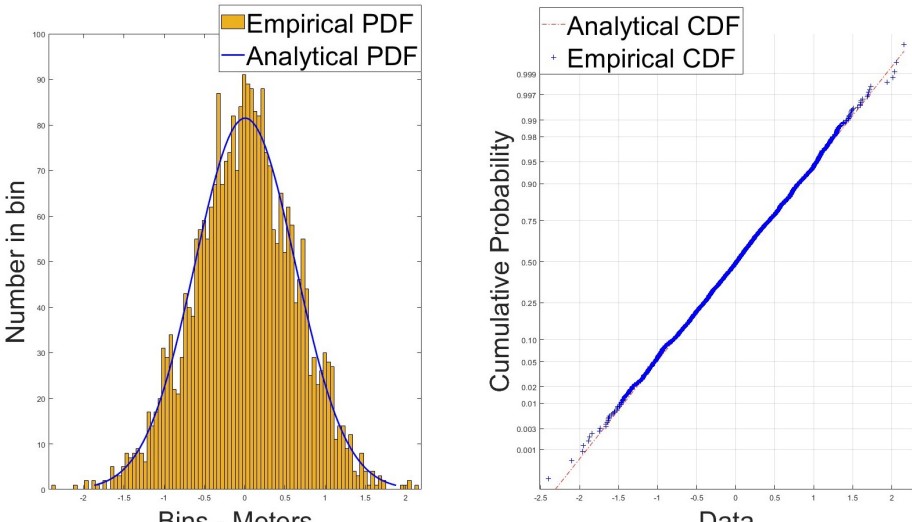

**Figure 8.** CMS measurement distribution—PRN 12.

**Table 3.** Expected and calculated means and variances—CMS distribution—no atmospheric errors.

| PRN | C/No (dB-Hz) | Expected Mean (m) | Measured Mean (m) | Expected Std Dev (m) | Measured Std Dev (m) |
|-----|--------------|-------------------|-------------------|----------------------|----------------------|
| 24 | 45.1 | 0 | −0.0254 | 0.588 | 0.64 |
| 8 | 45 | 0 | 0.0154 | 0.5929 | 0.6184 |
| 12 | 45.5 | 0 | 0.0056 | 0.5596 | 0.6250 |
| 2 | 45.3 | 0 | 0.0164 | 0.5735 | 0.6806 |
| 11 | 45.3 | 0 | 0.0668 | 0.5772 | 0.6625 |
| 25 | 41.96 | 0 | 0.0692 | 0.8489 | 0.7908 |

Table 3 shows the measured means are not zero, as expected, and tend to be in the cm region. This is likely due to the fact that in theory the delay estimates calculated in the DLL and SLL should be equal; however, in practice they tend to be slightly different, due to various factors including front-end filtering [28]. The measured standard deviations, as expected, also differ from those calculated. This is due to neglecting the covariance term in Equation (16). The difference in calculated and measured deviations vary from satellite to satellite, indicating that for precision applications this would need to be calculated for each visible PRN. Averaging the five measurements where measured deviations were higher than calculated suggests that cov(DLL,SLL) is approximately 0.0335 m; however, results are too sparse and vary too wildly for this result to be meaningful. Therefore, approximating the distribution as zero mean Gaussian distributed with standard deviation according to Equation (16) would not be suitable for applications such as Precise Point Positioning (PPP); however, it seems to be satisfactory for the determination of a metric for multipath detection. Obviously the metric would be much more sensitive if actual means and variances are used, however this is not necessarily possible for the average stand-alone receiver and so the measured (empirically derived) distributions are plotted as histograms against a normal distribution with expected characteristics (analytically derived) for the visual inspection of the estimated distribution fit.

In Figures 9 and 10 the data are plotted as a histogram and two normal distributions are then fitted to said histograms. The red curve is the analytically derived PDF, centred at zero (the expected mean), and the blue curve is the analytically derived PDF, centred at the measured means. It can be seen that approximating the distribution as zero mean has very little effect as both green and blue curves appear to approximately fit to the data.

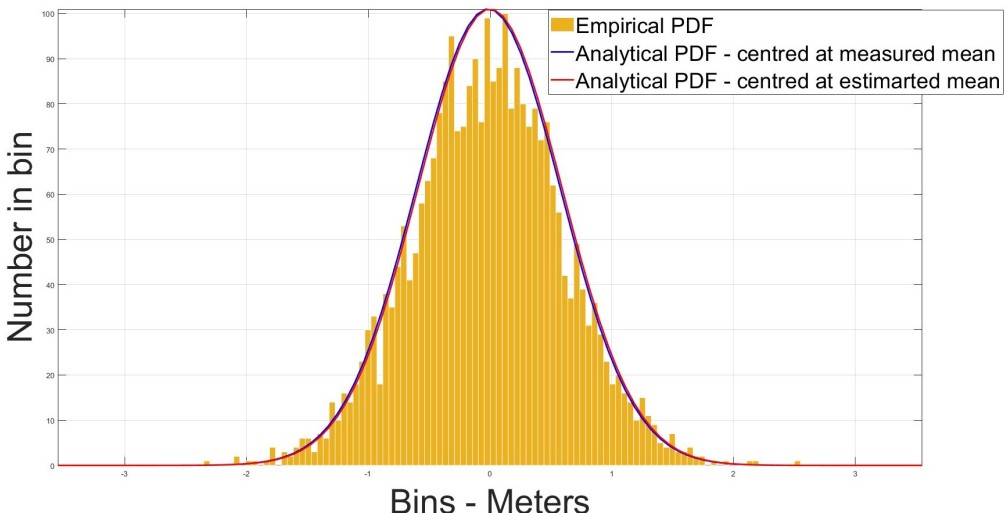

**Figure 9.** Empirical vs. analytical distribution—PRN 8.

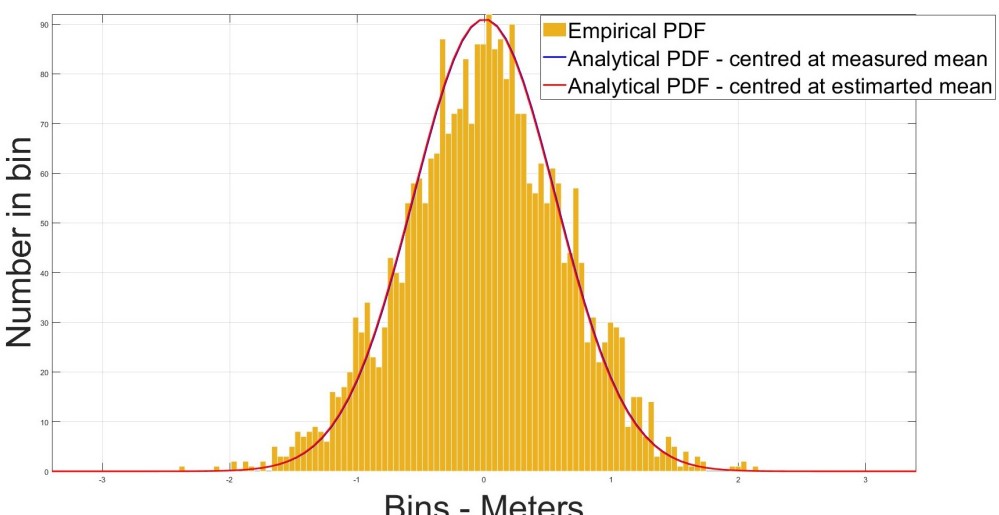

**Figure 10.** Empirical vs. analytical distribution—PRN 12.

In Equation (11) there is no ionospheric delay term, which sets the CMS measurement apart from the CMC measurement, as it is this that prevents single-frequency, single-point users from accurately using CMC to detect/measure multipath. The simulation scenario was repeated, with atmospheric errors turned on. Ionospheric errors are modelled using the Klobuchar Model [29] and tropospheric errors are modelled using the Saastemoinen model [30]. All other parameters including those set in the SDR are equal.

From Table 4 it can be seen that the means are shifted by a few cm and the deviations differ by millimeters (mm). This suggests that the distribution of the measurement is unaffected by the atmospheric errors and the ionospheric term present in both code and subcarrier measurements is cancelled when differencing the two. To visually assess the impact of atmospheric errors, the distributions with and without atmospheric errors for two satellites are plotted side-by-side in Figures 11–14.

**Table 4.** SDR parameters for measurement distribution experiments.

| PRN | C/No (dB-Hz) | Measured Mean (m) | Diff (m) | Measured Std Dev (m) | Diff (m) |
|-----|--------------|-------------------|----------|----------------------|----------|
| 11 | 45.3 | 0.0480 | 0.0226 | 0.6636 | 0.0011 |
| 8 | 45.0 | 00266 | 0.042 | 0.6118 | 0.006 |
| 12 | 45.5 | 0.414 | 0.0358 | 0.6327 | 0.0077 |
| 24 | 45.2 | −0.0049 | 0.0205 | 0.6207 | 0.0193 |
| 2 | 45.3 | 0.0087 | 0.0077 | 0.6337 | 0.469 |
| 25 | 41.97 | −0.0840 | 0.0148 | 0.7767 | 0.0141 |

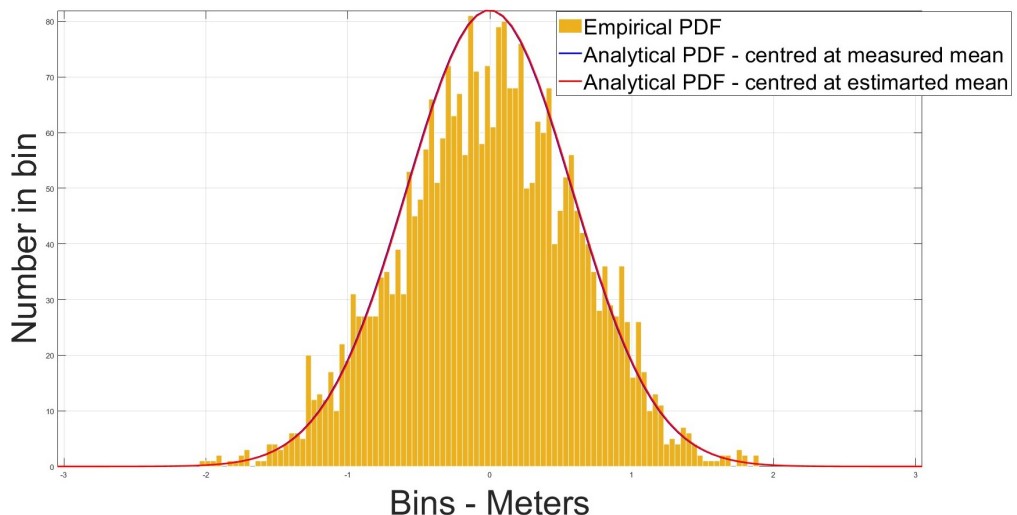

**Figure 11.** CMS distribution—with atmos errors—PRN 24.

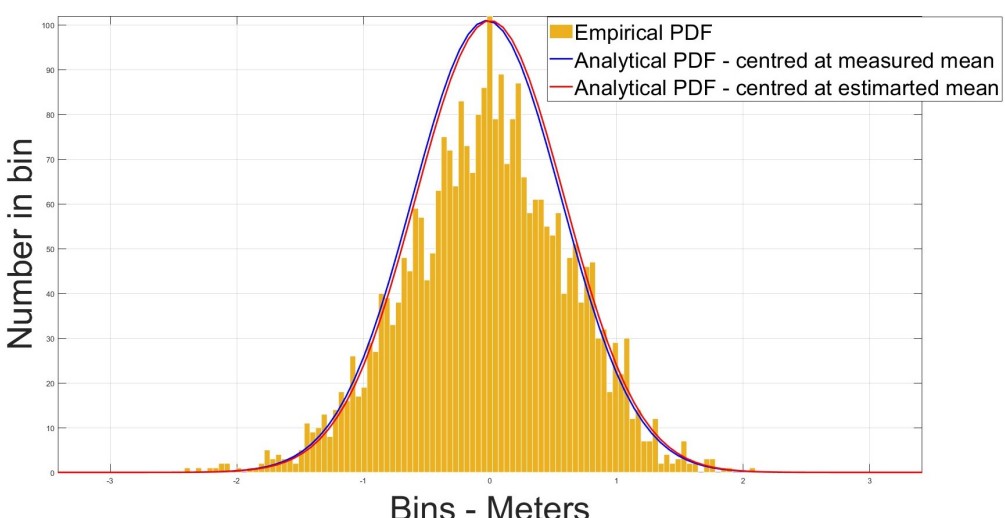

**Figure 12.** CMS distribution—no atmos errors—PRN 24.

It is clear from Figures 11–14 that assuming the CMS measurement to be zero mean Gaussian distributed with standard deviation calculated according to Equation (16), although not accurate enough for high precision applications, is a good approximation for the actual distribution. This will be used to define a threshold value for multipath detection. As the measurement is defined by the noise from the two loops under no multipath conditions, any value over the threshold would indicate the presence of multipath. The threshold will be set using Equation (16), which uses receiver parameters and signal carrier to noise. Thus

C/No must be monitored to ensure an adaptive threshold that responds to changes in signal power.

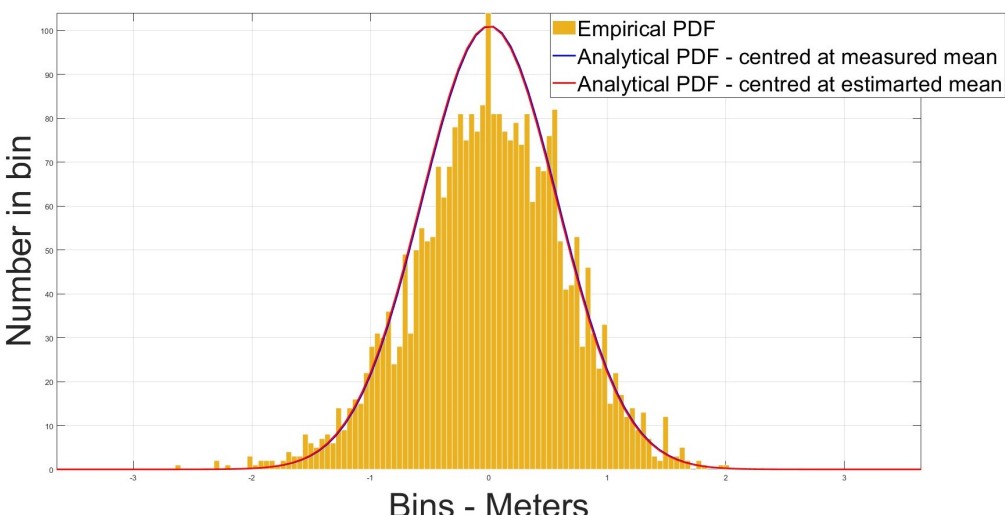

**Figure 13.** CMS distribution—with atmos errors—PRN 2.

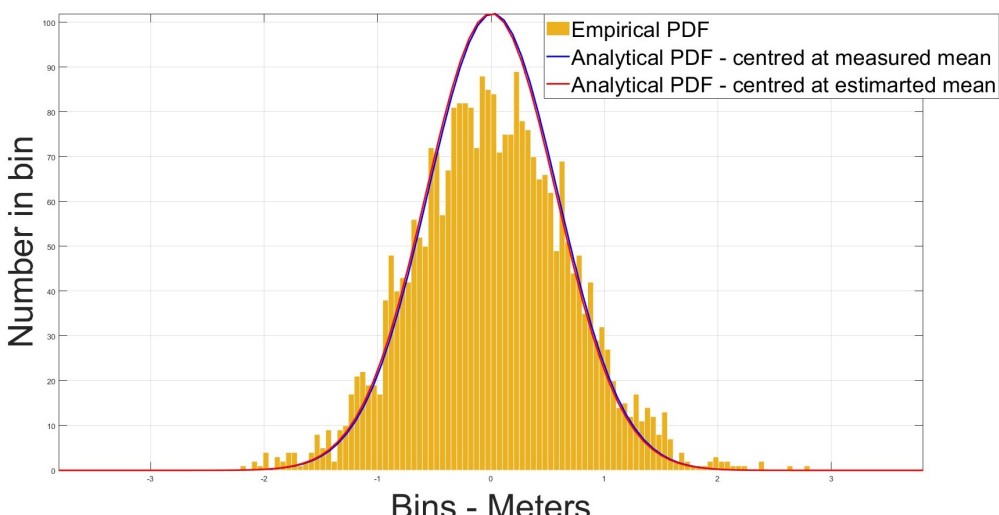

**Figure 14.** CMS distribution—no atmos errors—PRN 2.

### *3.2. Threshold Calculation*

It was shown above that under no-multipath conditions, the CMS measurement can be approximated as zero mean Gaussian distributed with a standard deviation, $\sigma$, according to Equation (16). Thus, providing the discriminator implementation within the receiver is known, the noise in the code, $\sigma_{tDLL}$, and subcarrier loops, $\sigma_{tSLL}$, and the resulting CMS measurement, $\sigma_{tCMS}$, can be estimated. This estimate will be continually updated by the receiver monitoring C/No and will be used to set an adaptive threshold according to Equation (17).

$$thres_{cms} = \pm m_{exp}\sigma_{t_{CMS}} \qquad (17)$$

where $m_{exp}$ is a threshold expansion factor that determines the false-alarm rate of the multipath detection metric. Figure 15 show the effect of increasing $m_{exp}$ from 1 to 3. Approximately 68.27% of measurements are expected to fall within $1\sigma$, 95.45% within $2\sigma$ and 99.73% within $3\sigma$. Thus, with a $m_{exp} = 3$, the probability of false alarm, $p_{fa}$, would be 0.27%, and for every 1000 measurements approximately 3 would be expected to exceed the threshold and cause a false alarm. This seems too high, especially when compared to a

five-sigma threshold which would result in a 0.00006% chance of false alarm and seems much more reasonable. It is clear that threshold selection is a trade-off between sensitivity (i.e., probability of detection $p_d$) and false alarm rate, $p_{fa}$; a high threshold value would ensure a low $p_{fa}$; however, a larger CMS value is required before multipath is detected and a low threshold would improve the detection rate at the expense of more false alarms. A list of threshold expansion factors and their associated probability of false alarm can be seen in Table 5.

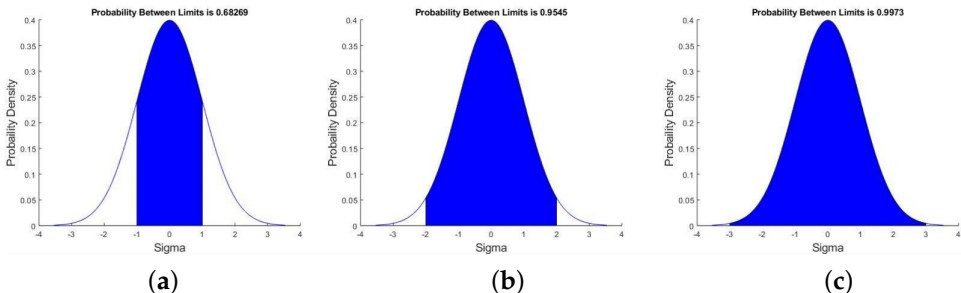

|  | (a) | (b) | (c) |

**Figure 15.** Threshold Determination: (**a**) $m_{exp}$ = 1, (**b**) $m_{exp}$ = 2, (**c**) $m_{exp}$ = 3.

**Table 5.** Expansion factors and respective probabilities of false alarm.

| $m_{exp}$ | 1 | 2 | 3 | 4 | 5 | 6 |
|---|---|---|---|---|---|---|
| $p_{fa}$ | 0.3174 | 0.0456 | 0.0027 | 0.000063 | 0.00000057 | 0.000000002 |
| $p_{fa}$% | 31.74 | 4.56 | 0.27 | 0.0063 | 0.000057 | 0.0000002 |

### 3.3. Performance Analysis

The performance of the CMS measurement as a multipath detection metric will be evaluated using Multipath Error Envelopes (MEE). A common simplification/assumption that the direct signal is always available and only one multipath echo is present is made at this point. Typically, two paths, in-phase (zero-degree phase shift) and out-of-phase (180 degree phase shift), are assumed to be present and the relative delay of the multipath signal is swept through a range of values (0–1.5 Chips). The code multipath errors are then expressed as a function of multipath delay forming the MEE. MEEs will be used to demonstrate the metric's sensitivity to various signal and receiver parameters, which include:

- Geometric path delay;
- Multipath amplitude;
- Chip spacing between tracking correlators;
- Front-end bandwidth.

For the purpose of simplifying signal generation, the Galileo E1B MBOC modulation will be simplified to BOC(1, 1), as this is the major component. The line-of-sight and in-/anti-phase multipath signals are generated in MATLAB using Equation (4), such that a subcarrier square wave with frequency $f_s = 1.023$ MHz is modulated onto a Galileo PRN 1 spreading code of length 4092 chips with a frequency $f_c = 1.023$ MHz, and the two are subsequently modulated onto a carrier waveform at intermediate frequency (IF) of 14.58 MHz and sampled at 53 MHz. This is representative of how GNSS signals would arrive at the digital signal processing (DSP) section of the receiver. The amplitude of the multipath echo is determined by $\alpha$ in Equation (4) and typically takes the value 0.5 to represent the worst-case reflection amplitude; however, values of 0.1 and 0.3 will also be used here to assess the measurements sensitivity to echo amplitude. Multipath delay is incremented every 4 s from 0 to 1.5 chips, as multipath range errors stop being significant at this point [31]. Signals are tracked using a simplified version of SDR described in Section 2, operating solely in PLL mode. Receiver parameters that will stay unchanged throughout the experiments can be seen in Table 6.

**Table 6.** SDR Parameters for measurement sensitivity experiments.

| Receiver Parameter | Set Value |
|---|---|
| Sampling Frequency | 53 MHz |
| PLL Noise Bandwidth | 15 Hz |
| DLL Noise Bandwidth | 0.5 Hz |
| SLL Noise Bandwidth | 0.5 Hz |
| PLL Discriminator | 2 Quadrant Arc-tangent |
| DLL Discriminator | Early Minus Late Power |
| SLL Discriminator | Early Minus Late Power |
| Tracking Correlation Time | 4 ms |

To assess the impact of not only correlator spacing but relative correlator spacing differences, multiple scenarios will be analysed. The code and subcarrier correlation functions can be seen in Figure 16, where marked points represent the chosen discriminator spacings used for the scenarios listed in Table 7. The equivalent equal spacing scenarios represent typical operating set-ups, the wide scenario has typical spacings for tracking-only purposes, and the narrow scenario represents nEML spacings used for multipath mitigation. The wide–narrow and wide–very narrow scenarios have been included as they represent, assumed, optimal spacings for the CMS measurement.

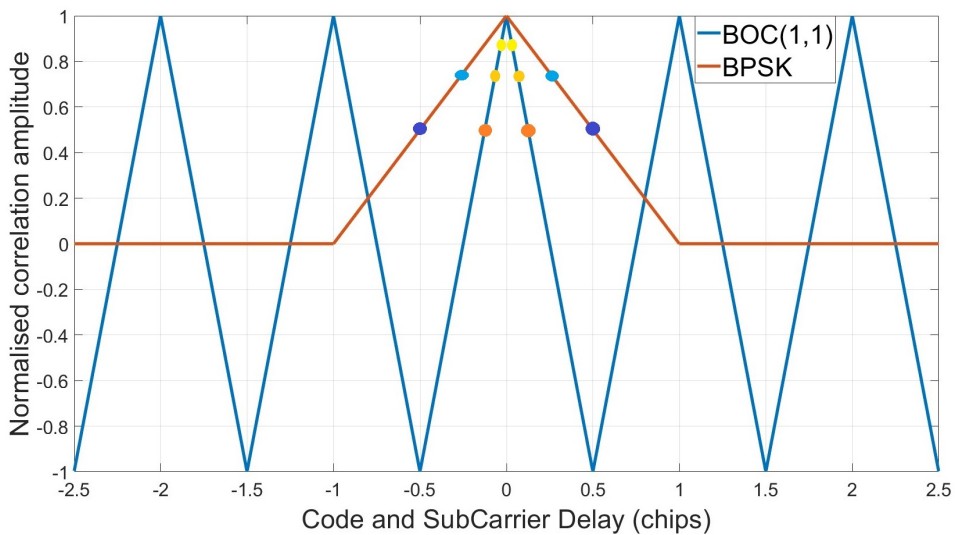

**Figure 16.** Code and subcarrier correlator spacings.

**Table 7.** Discriminator Spacing Scenarios.

| Scenario Description | Scenario Number | DLL E-L Spacing | SLL E-L Spacing |
|---|---|---|---|
| Equivalent Equal Spacing—Wide | 1 | 1 (Dark Blue) | 0.25 (Dark Orange) |
| Wide—Narrow | 2 | 1 (Dark Blue) | 0.125 (Light Orange) |
| Wide—Very Narrow | 3 | 1 (Dark Blue) | 0.1 (Yellow) |
| Equivalent Equal Spacing—Narrow | 4 | 0.25 (Light Blue) | 0.125 (Light Orange) |

Multipath detection techniques are typically assessed by their sensitivity and effectiveness. Both qualities are defined in [4]. Sensitivity is defined by the magnitude of the metric Variation Profile (VP). If the CMS measurement exceeds a given threshold, multipath is detected and the method is deemed to be sensitive. Threshold values have been included in the VPs to assess sensitivity. Techniques are classed as effective if they are sensitive at points where significant multipath ranging errors exist. Thus, measurement Variation Profiles will be presented to assess sensitivity, in conjunction with Multipath Error Envelopes

to assess the effectiveness. As three of the four scenarios use the same Code Correlator Spacing, the MEE for the code loop with an E-L spacing of 1 chip can be seen in Figure 17. The multipath is incremented every 4 s and a moving average filter 4 s long is used for smoothing purposes in both MEE and VP generation. This four-second moving average filter is employed throughout this paper for the CMS measurements in all experiments.

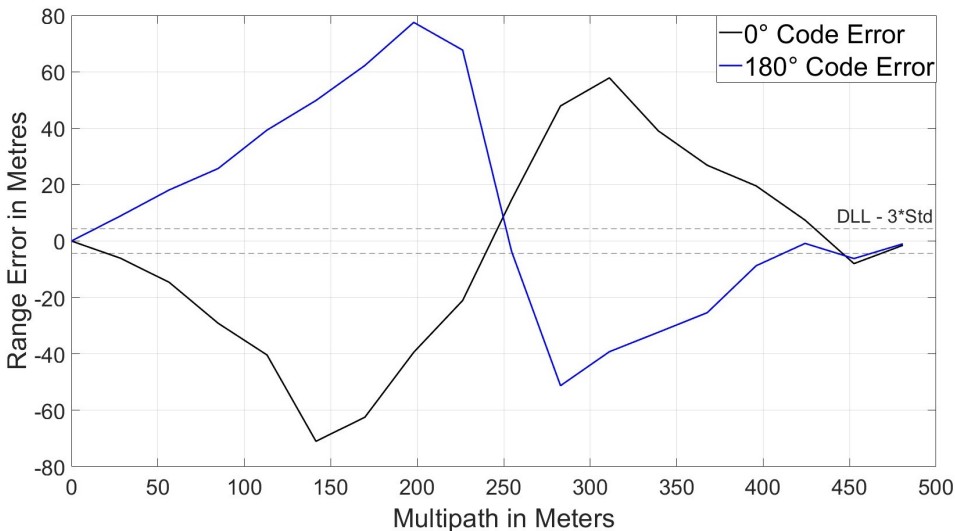

**Figure 17.** MEE for code loop—E-L spacing 1 chip.

The VP for the equivalent equal spacing scenario can be seen in Figure 18. It can be seen that the Code MEE and the CMS VP are similar. This is because ranging errors on the code are much larger than that on the subcarrier and dominate the measurement. The ME for the subcarrier with 0.25 chip E-L spacing (scenario 1) can be seen in Figure 19.

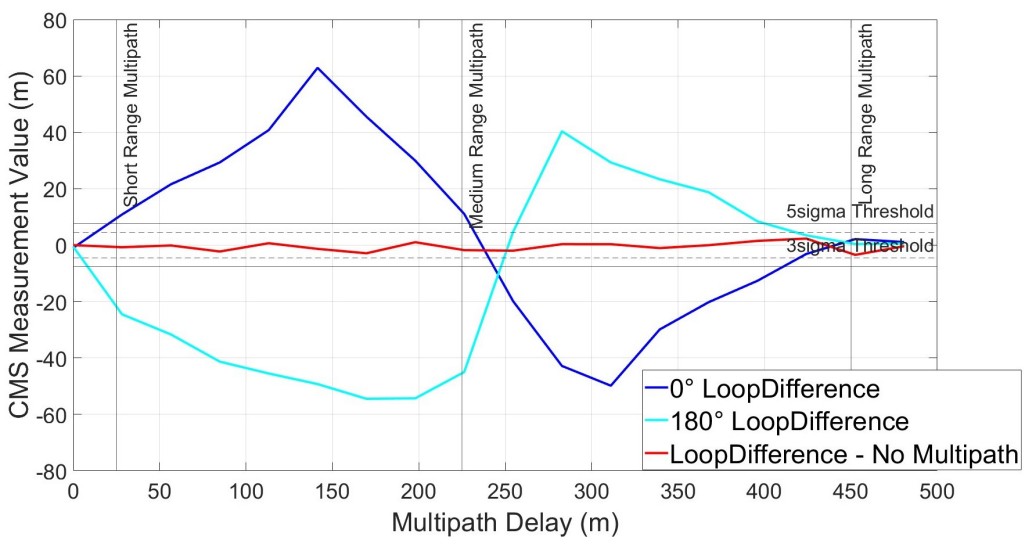

**Figure 18.** CMS Variation Profile—equal spacing—wide.

It can be seen in Figure 19 the subcarrier MEE profile differs from that of the code and therefore will alternate between adding constructively and destructively within the CMS measurement. Hence, the VP and ME are similar, not equal. However, it can be seen that when the measurement is sensitive (i.e., measurement exceeds the threshold) it is always effective. Three regions can be seen in Figure 18. These relate to the multipath delays and are classified as either:

- Short-range < 25 m;

- Medium-range 25 m < 225 m;
- Long-range > 225 m.

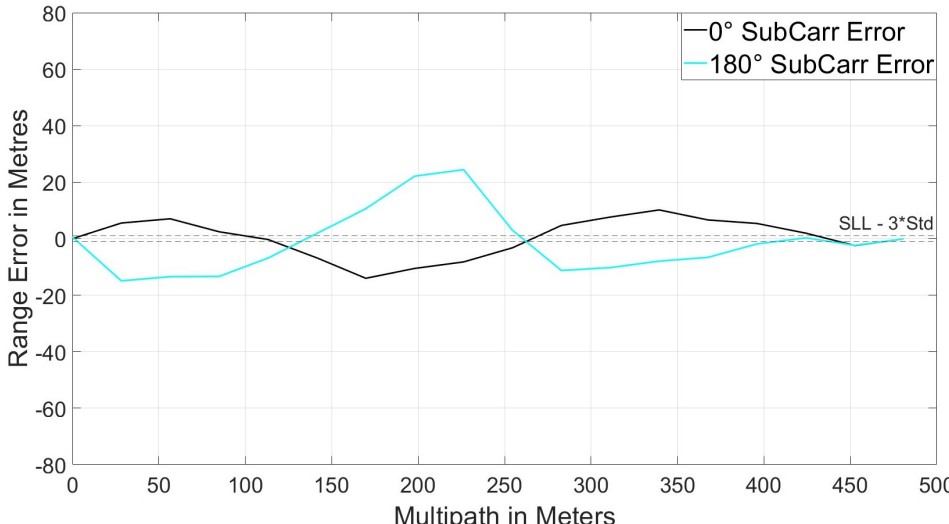

**Figure 19.** MEE for subcarrier loop—E-L spacing 0.25 chip.

Short-range multipath is notoriously difficult to detect and, to the best of the authors' knowledge, current SQM methods cannot detect multipath signals with a delay of less than 25 m. In Figure 18, the CMS value crosses the threshold, indicating the presence of multipath at 20 m. Medium-range multipath is detected across the whole range. Between 225 m and 260 m, multipath is not detected; however, this corresponds to the region in the code MEE where the multipath error is transitioning from positive to negative and no significant multipath error exists. Aside from this region, long-range multipath is detected.

To isolate the code multipath in the CMS equation, the correlator spacing in the SLL has been reduced. This serves two purposes: firstly, it reduces the noise in the SLL measurement, as is evident from Equation (14). Second, reducing the correlator spacing has been shown to mitigate the effects of multipath [9]. If the subcarrier noise and multipath are reduced significantly, the CMS equation can be approximated as Equation (18). The subcarrier MEEs for scenarios 2 and 3 can be seen in Figures 20 and 21, which demonstrate that reducing the subcarrier correlator spacing also reduces the multipath error in the SLL.

$$\text{CMS} = MP_{i,k}^{s} + \epsilon_{i,k}^{s} \tag{18}$$

VPs for scenarios 2 and 3 can be seen in Figures 22 and 23. There is very little difference between these and the VP for scenario 1 in Figure 18. This indicates that reducing the subcarrier multipath has little effect on CMS as a multipath detection metric, however it would certainly improve the CMS measurement if used for measuring multipath.

In the final scenario, equivalent correlator spacings are used again, except both code and subcarrier are located much closer to their respective peaks compared to scenario 1. This configuration is traditionally used for the purpose of multipath mitigation, as demonstrated above. Figure 24 shows the VP for scenario 4.

The MEE profile is different compared to the other three scenarios. Short-range multipath is no longer detectable and only the longer end of medium-range and the shorter end of long-range multipath delays can be detected. If we look at the code loop MEE for this scenario in Figure 25, it can be seen that the measurement remains effective when sensitive, thus detecting significant multipath ranging errors.

To determine the measurements' sensitivity to multipath echo amplitude, the receiver is returned to scenario 1 configuration and multipath signals of varying amplitude are be processed as before. Amplitudes take three values—0.5, 0.3 and 0.1—relative to the LOS component, and the VPs are plotted in Figure 26.

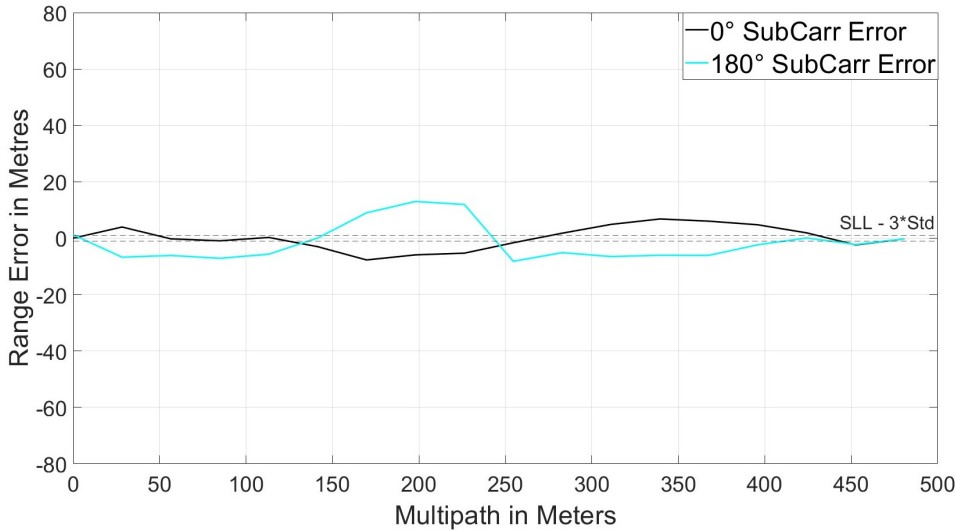

**Figure 20.** Subcarrier Multipath Error Envelopes—narrow spacing.

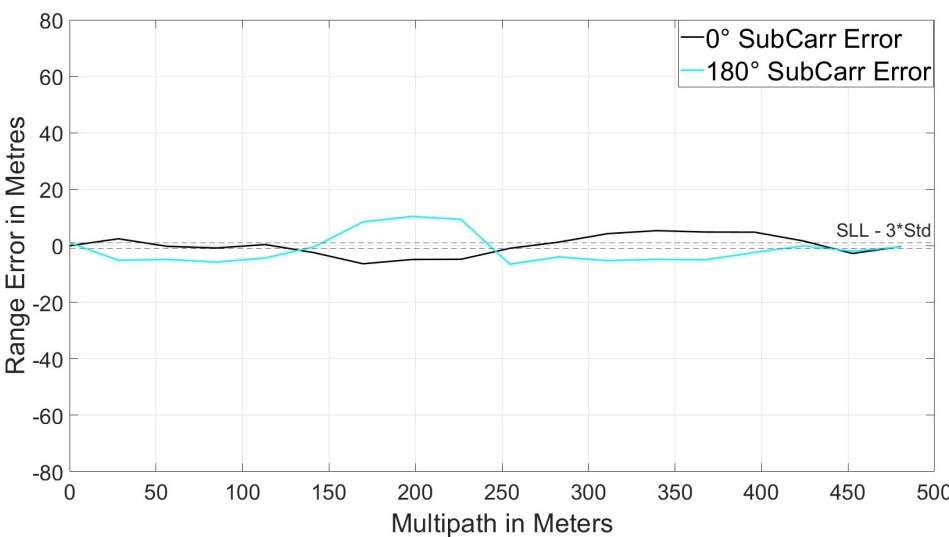

**Figure 21.** Subcarrier.

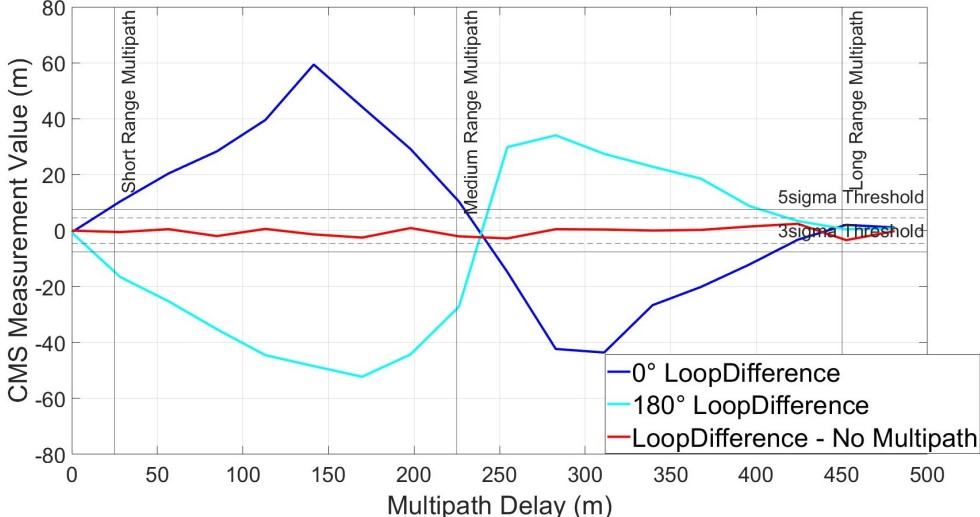

**Figure 22.** CMS Variation Profile—wide–narrow spacing.

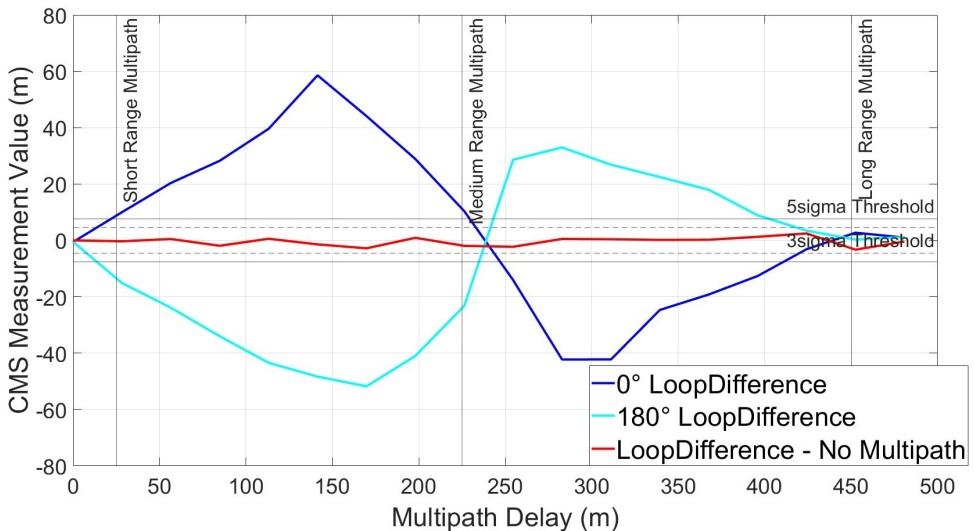

**Figure 23.** CMS Variation Profile—wide–very narrow spacing.

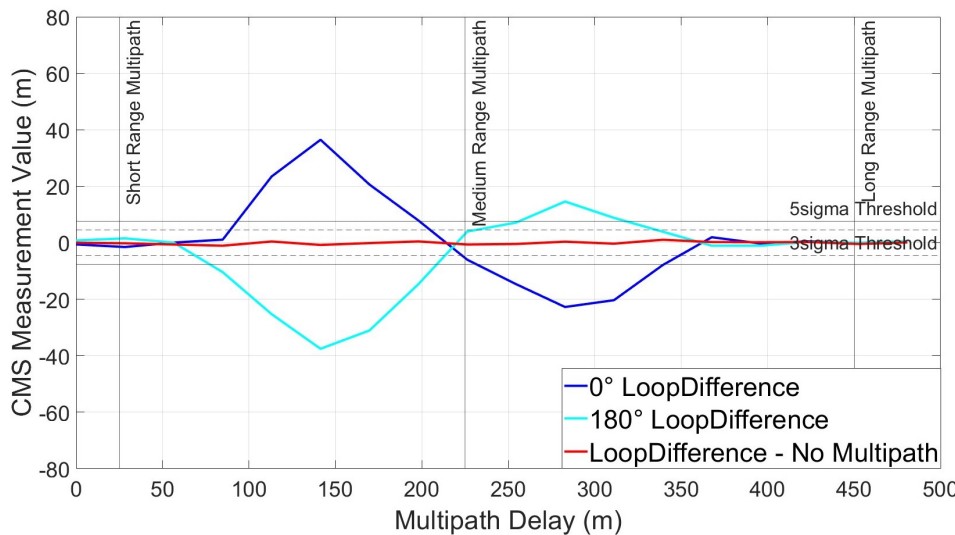

**Figure 24.** CMS Variation Profile—equal spacing—narrow.

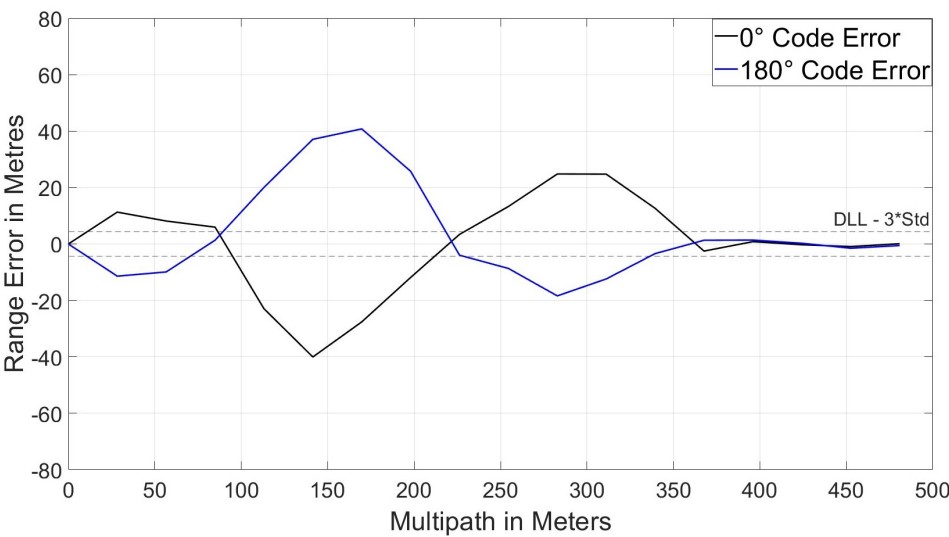

**Figure 25.** MEE for code loop—E-L spacing 0.5 chip.

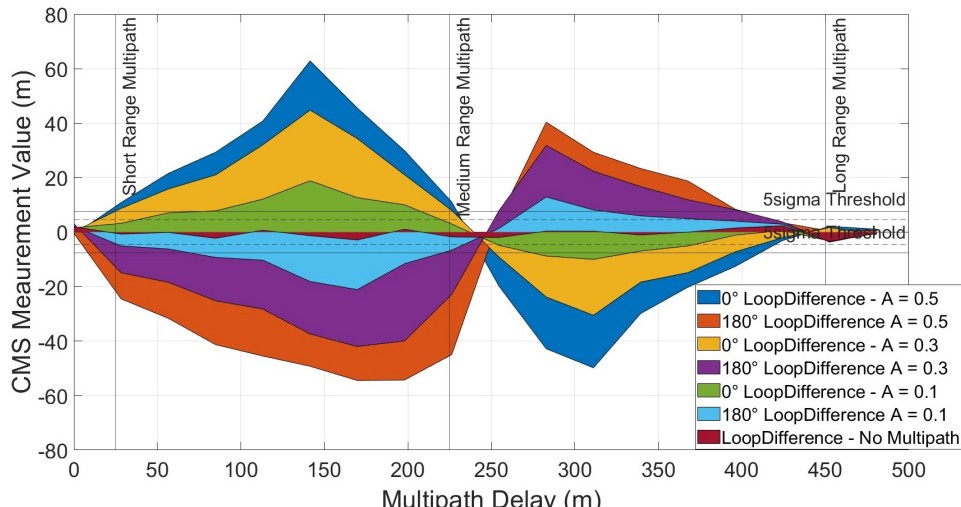

**Figure 26.** CMS Variation Profile—equal spacing—wide—amplitude sensitivity.

The amplitude seems to determine the shortest delay detectable: when $\alpha = 0.5$, delays of less than 25 m are detected; when $\alpha = 0.3$, delays of 25 m or greater are detectable; and when $\alpha = 0.1$, delays greater than 50 m are detected. Also, the region at approximately 250 m meters where there is no multipath detected becomes wider as the multipath amplitude decreases. However, just like above, the CMS measurements are always effective when sensitive and multipath delays that are undetectable at low amplitudes cause less significant ranging errors.

To determine the measurement's sensitivity to front-end bandwidth, the simulation consisting of six Galileo satellites (described above) was processed using the RF output of the Skydel GSG-8 and an NT1065 front-end with a bandwidth of 53 MHz. The NT1065 down-converts the RF signal to an intermediate frequency (IF) and the from analogue to digital format which can then be processed through an SDR. The scenario parameters were exactly the same as those in Table 1, except atmospheric errors models are applied, as described in Section 3, and single-fixed offset multipath echos with a delay of 75 m were added to two signals at 40 s into the file. The SDR parameters also match those in Table 6. However, an intermediate frequency of 14.58 MHz is used, as the front-end does not down-convert to base band. Figures 27 and 28 show the measured CMS values for a signal not experiencing multipath at all and one where the multipath echo is turned on at 40 s.

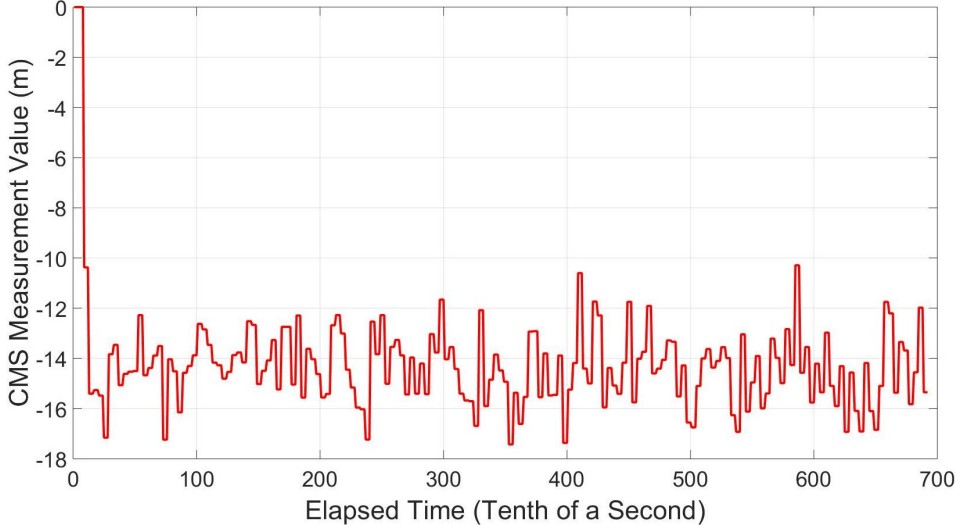

**Figure 27.** CMS measurement—no multipath—front-end sensitivity.

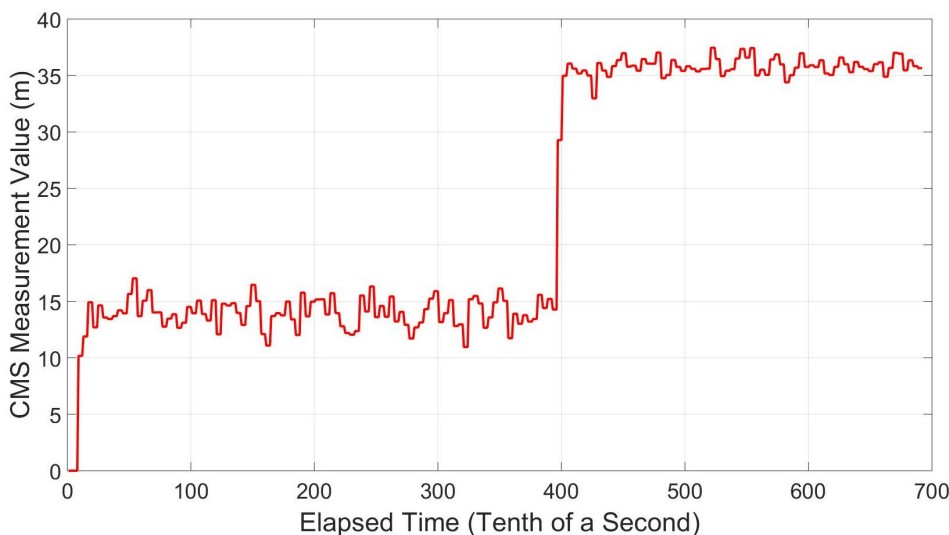

**Figure 28.** CMS measurement—75 m multipath echo at 40 s—front-end sensitivity.

In Figures 27 and 28, it can be seen that even under no-multipath conditions there is a constant bias in the CMS value. This is likely caused by the profile of the front-end filter, which is far from flat, causing slight distortion in the two loops, inducing an error.

A correlation function monitor was used to characterise the code correlation slope with a clean signal chosen for examination. The result is averaged over 800 code periods to obtain the averaged correlation function (green slope) in Figure 29.

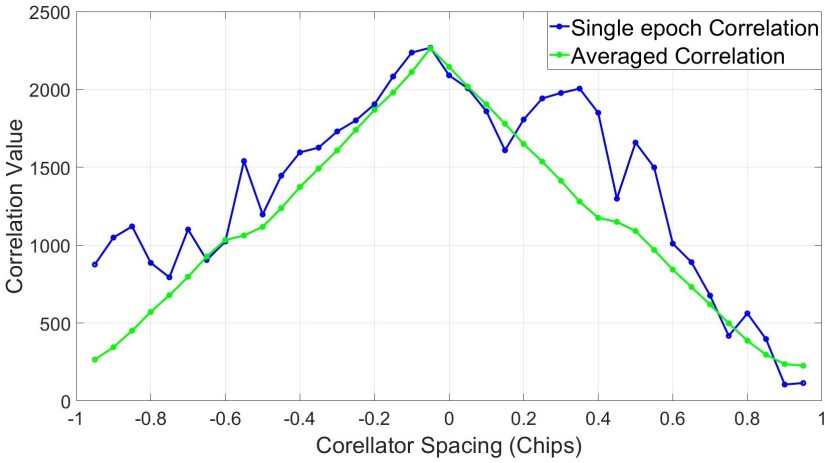

**Figure 29.** Correlation function—front-end band-limiting.

Figure 29 shows the slopes of the correlation function are distorted to the point where the peak no longer lies at zero chips. To determine the effect this distortion has on the two loops, the discriminator is mapped by taking sequential early minus late measurements along the correlation function. With no distortion, the crossing point of this would lie at zero (in the x axis); however, the front-end filtering has caused a skew in the discriminator, moving this to $-0.05$ chips, as can be seen in Figure 30.

When converted from chips to meters using Equation (19), the skew in the discriminator gives a value of approximately 14.9 m which is the value of the bias seen in Figures 27 and 28. This is approximately the value of the averaged difference between the two loops in a clean environment for the NT1065 front-end operating with a front end bandwidth of 53 MHZ. This demonstrates the measurement's sensitivity to the front-end filter and therefore must be calibrated for before use.

$$Value_{meters} = (Value_{chips}) * c/fc \tag{19}$$

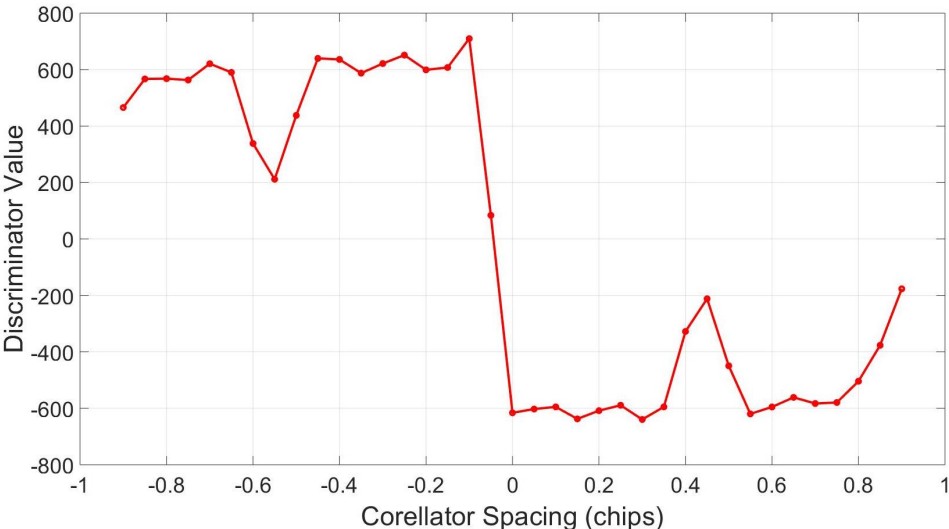

**Figure 30.** Discriminator function—front-end band-limiting.

## 4. Discussion

In Section 3, it was shown that the CMS measurement is sensitive to multipath signal amplitude and front-end band-limiting. In scenarios 1, 2 and 3 it was shown that all ranges of multipath that cause significant ranging error can be detected through the monitoring of the CMS measurement value and comparison with an adaptive threshold. This is true when the receiver is operating with a wide code loop correlator spacing and an equivalent-equal or narrow subcarrier loop correlator spacing. The CMS measurement was not as effective when a narrow correlator is employed in the DLL, as this mitigates the multipath. To verify this, single-echo fixed-offset multipath scenarios were simulated using the GSG-8 simulator. The scenario parameters were exactly the same as those in Table 1; however, atmospheric errors models are applied, as described in Section 3, and output as digital base-band to avoid the effects of front-end filtering. The SDR parameters also match those in Table 2. Only the one satellite signal was simulated and the multipath echo is enabled 10 s into the file; its phase and frequency are equal to that of the LOS and its amplitude is half that of LOS. The DLL E-L spacing = 1 chip and the SLL E-L spacing = 0.125 chips (scenario 3 in Section 3). The various delays tested include:

- Short-range, 20–25 m;
- Medium-range, 150 m & 250 m;
- Long-range, 375 m.

It was found that the smallest detectable Multipath Delay was 21 m. Figure 31 shows the CMS value crossing the threshold almost instantaneously, as the multipath echo is injected at 10 s. Outcomes were positive for all ranges under test, showing no false alarms while no multipath is present and instantaneous alarms sounded once the echo was injected. Results can be seen in Figures 31–35.

It can be seen from Figures 31–35 that, even when using the higher metric threshold value of $5\sigma_{metric}$, multipath echos greater than 21 m can be detected via the monitoring of the CMS measurement in conjunction with a C/No monitor.

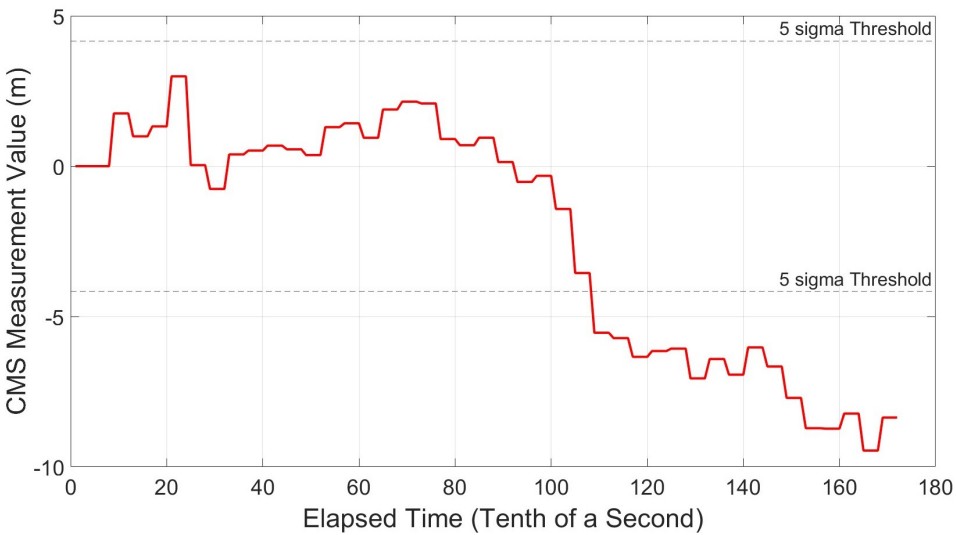

**Figure 31.** CMS Variation Profile—equal spacing—narrow—multipath delay = 21 m.

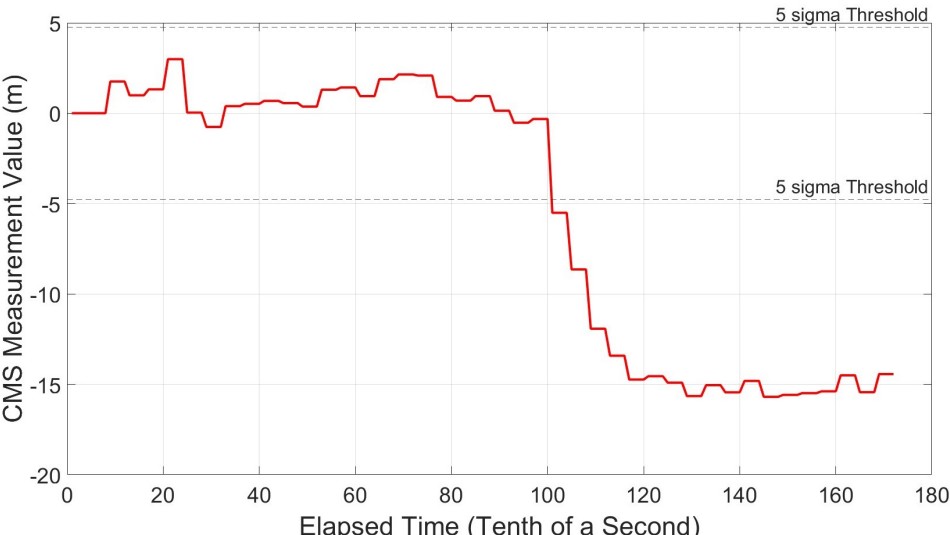

**Figure 32.** CMS Variation Profile—equal spacing—narrow—multipath delay = 25 m.

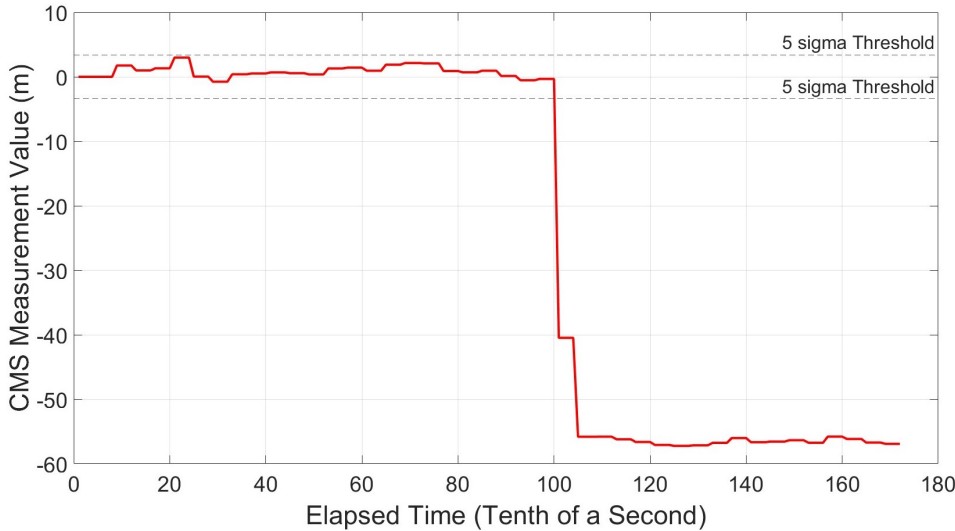

**Figure 33.** CMS Variation Profile—equal spacing—narrow—multipath delay = 150 m.

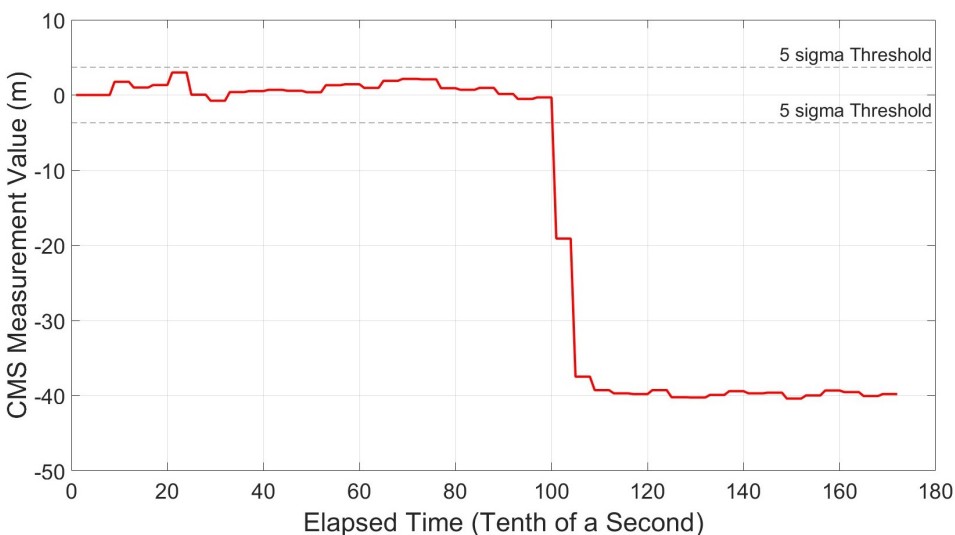

**Figure 34.** CMS Variation Profile—equal spacing—narrow—multipath delay Delay = 225 m.

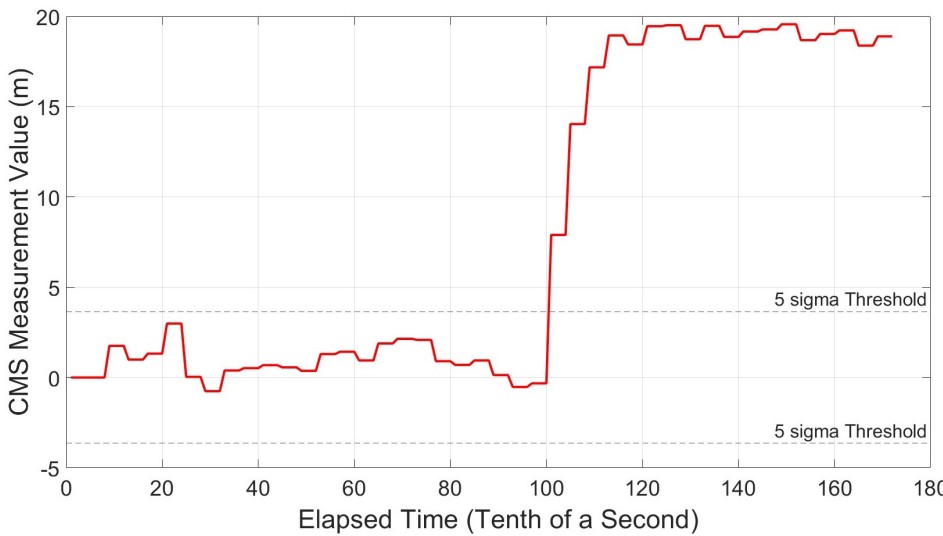

**Figure 35.** CMS Variation Profile—equal spacing—narrow—multipath delay = 375 m.

## 5. Conclusions

This paper has presented a novel single-difference measurement, Code Minus Sub-carrier (CMS), for the purpose of multipath detection. The measurement has been defined mathematically, and through the characterisation of the measurement distribution under no-multipath conditions, a threshold for multipath detection was calculated. The threshold can be estimated using expected noise values of the two control loops or measured through a calibration campaign. The estimated value from the expected loop noise (calculated from the measured C/No) provides an overestimate and is thus acceptable to use as a threshold; calibration would allow for a more sensitive measurement. The threshold is adaptive and requires the carrier-to-noise of each tracked signal to be monitored, which exists in the vast majority of receivers and so the additional hardware required to generate this measurement is kept minimal, as is any additional computational complexity. Measurements falling within threshold bounds are determined to not be suffering from multipath. Once the threshold has been surpassed, multipath is detected. This flag can be used within a fault detection and exclusion algorithm such as Receiver Autonomous Integrity Monitoring (RAIM) schemes. RAIM detects faults using redundant GPS pseudorange measurements to ensure all psuedoranges are consistent with the computed position. Any satellite pseu-dorange measurement that differs from the expected value is excluded from the navigation

solution. In much the same sense, any satellite signal with a raised multipath flag could also be excluded.

Multipath Error Envelopes and measurement Variation Profiles demonstrated the measurement's sensitivity and effectiveness for a selection of code and subcarrier correlator spacings and various multipath amplitudes. By minimising the effect of multipath in the subcarrier through use of narrow correlators in the SLL and a wide correlator spacing in the DLL (optimal configuration), previously undetectable multipath delays of less than 25 m were shown to be detectable. When both DLL and SLL utilise narrow correlator spacings, for multipath mitigation, the ranges of detectable multipath reduce significantly. However, this is due to the multipath being suppressed and any echoes causing significant (larger than the threshold) ranging errors are detected. Short-, medium- and long-range multipath delays were all detected when the optimal correlator spacing (DLL - wide, SLL - Narrow) is chosen. As the amplitude of the multipath signal was reduced, so too was the resulting ranging error, which in turn determined the shortest detectable delay. It was also shown that the front-end band-limiting of the signal causes a frequency-dependant bias in the measurement that needs to be calibrated for and removed. Through simulations, the CMS measurement was shown to be an effective multipath detection metric at all ranges where it was deemed to be sensitive. This method will work as a useful multipath detection metric for single-point, stand-alone receivers.

Further work is planned involving testing CMS's performance as a multipath detection metric in a live sky environment and as a metric for spoofing detection. Live sky verification will also include a discussion on suitable averaging times. Through this study it was shown that the measurement VP was similar to the code loop MEE and, so, research into the measurement's ability to measure and thus correct multipath error will also be performed. This would enable stand-alone receivers to correct their measurements and obtain precision not previously afforded to them.

**Author Contributions:** Conceptualisation, M.A. and P.B.; methodology, M.A. and P.B.; software, M.A. and P.B.; validation, M.A. and P.B.; formal analysis, M.A.; investigation, M.A. and P.B.; resources, M.A. and P.B.; data curation, M.A.; writing—original draft preparation, M.A.; writing—review and editing, M.A.; visualisation, M.A.; supervision, P.B.; project administration, M.A. and P.B.; funding acquisition, M.A. and P.B. All authors have read and agreed to the published version of the manuscript.

**Funding:** This research was funded by EPSRC, grant number EP/R513283/1.

**Institutional Review Board Statement:** Not applicable.

**Informed Consent Statement:** Not applicable.

**Data Availability Statement:** Data available on request from the authors.

**Acknowledgments:** I would like to thank my research institute, The Nottingham Geospatial Institute, for giving me the opportunity to undertake this research. The largest thanks go to my Mother and Father though, as without their constant and unwavering support none of this would be possible.

**Conflicts of Interest:** The authors declare no conflict of interest.

## Abbreviations

The following abbreviations are used in this manuscript:

| | |
|---|---|
| ASPECT | Auto-correlation Side-Peak Cancellation Technique |
| BPSK | Binary Phase Sift Keying |
| BOC | Binary Offset Carrier |
| CDF | Cumulative Distribution Function |
| CMC | Code Minus Carrier |
| CMS | Code Minus Subcarrier |
| CNo | Carrier to Noise Ratio |
| DE | Double Estimator |
| DLL | Delayed Lock Loop |

| | |
|---|---|
| DPE | Double Phase Estimator |
| DSP | Digital Signal Processing |
| EML | Early-Minus-Late |
| FLL | Frequency Lock Loop |
| FPGA | Field Programmable Gate Arrays |
| GNSS | Global Navigation Satellite Systems |
| GPS | Global Positioning Service |
| HZA | High-Zenith Antenna |
| LoS | Line of Sight |
| MBOC | Multiplexed Binary Offset Carrier |
| MEE | Multipath Error Envelope |
| MLA | Multipath Limiting Antenna |
| PAC | Pulse Aperture Correlator |
| PDF | Probability Density Function |
| PLL | Phase Lock Loop |
| PNT | Position, Navigation and Timing |
| PPP | Precise Point Positioning |
| PRN | Pseudo Random Noise |
| PRS | Public Regulated Service |
| PSD | Power Spectral Density |
| RHCP | Right-Hand Circularly Polarised |
| RF | Radio Frequency |
| SACT | Subcarrier-Aided Code Tracking |
| SDR | Software Defined Receiver |
| SLL | Subcarrier Lock Loop |
| SoL | Safety of Life |
| SQM | Signal Quality Monitoring |
| VP | Variation Profile |

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
