# Peer review of "A Novel Single Differencing Measurement for Multipath Detection"

_remotesensing, doi:10.3390/rs15225312_

Round 1
Reviewer 1 Report
Comments and Suggestions for Authors
The study explored a novel single differencing measurement for multipath detection, the topic is interesting, and is well-written. The following concerns should be addressed before it goes to any future.
(1) For ABSTRACT, quantitative evaluation for the proposed method must be described, such as positioning accuracy and computational efficiency.
(2) The introduction to GNSS is to one-sided, why does it not include China’s Beidou?
(3) Are Figure 1 and 2 the results from other researchers? If yes, please give references. And Figures 1 and 2 lack the necessary explanation. What are the main points of these two figures?
(4) For METHODS, the corresponding flow chart should be added.
(5) Figures 14, 15, and 16 can be integrated into one figure.
(6) There are a lot of analysis and simulation in this paper, can you use the actual positioning data to verify it?
(7) The author indicated that “between 225 m and 260 m multipath is not detected”. Why? Is it because there is no multipath?
(8) There are some unscientific expression, such as MatLab, should be MATLAB.
Author Response
Dear Sir/Madam,
My I first say a very big thank you for taking the time to read and review our article. I understand that your time is limited and valuable and am very grateful that you have chosen to spend yours assisting us on this journey. I will address the points you have raised in bullet point form below. All changes to the manuscript have been highlighted in bold for your convenience.
- I have extended the abstract to discuss the computational efficiency and quantitative evaluation. Positioning accuracy is not relevant at this stage as the signals used within this work are simulated with known delays and in turn ranging errors. Future work including live sky testing with position accuracy tests will be addressed in this. (Work has already begun on this).
- I have included Beidou in the introduction.
- The figures are ours and we feel they are important as they explain how multipath effects the correlation functions which are the very things the measurement itself monitors.
- A flow chart has been added.
- Figures 14, 15 and 16 have been integrated into a single figure.
- Positioning accuracy is not relevant at this stage as the signals used within this work are simulated with known delays and in turn ranging errors. Future work including live sky testing with position accuracy tests will be addressed in this. (Work has already begun on this).
- Multipath ranging errors not occurring between 225 and 260m had already been addressed within the paper and the section has been highlighted.
- MatLab has been changed to MATLAB.
Reviewer 2 Report
Comments and Suggestions for Authors
This paper mainly focusses on a new single differencing technique for the accurate detection of multipath in standalone GNSS receivers receiving modernized Binary Offset Carrier (BOC) modulated signals.
The quality of the research is qualified and significant for understanding. The paper is recommended for publication after fixing some minor improvements of moderate revision. Notably, the paper needs to enhance the conclusion section, and general description on the previous research work should be added in.
Author Response
Dear Sir/Madam,
My I first say a very big thank you for taking the time to read and review our article. I understand that your time is limited and valuable and am very grateful that you have chosen to spend yours assisting us on this journey. I will address the points you have raised in bullet point form below. All changes to the manuscript have been highlighted in bold for your convenience.
- A general discussion of previous use of subcarrier aiding the code has been added to the introduction alongside the discussion of existing multipath detection techniques.
- The conclusion has been expanded and enhanced.
Reviewer 3 Report
Comments and Suggestions for Authors
Dear Authors,
very nice article and result presentation.
With kind regards,
Reviewer
Author Response
Dear Sir/Madam,
My I first say a very big thank you for taking the time to read and review our article. I understand that your time is limited and valuable and am very grateful that you have chosen to spend yours assisting us on this journey. I will address the points you have raised in bullet point form below.
Thank you for your kind words. We have worked hard on this paper and are extremely proud of it.
Reviewer 4 Report
Comments and Suggestions for Authors
Detailed remarks:
Line 236 - should be: "in relation" - not "in irelation".
Line 262 and beyond - it has not been clearly demonstrated why the covariance of DLL, SLL is small and negligible.
In my opinion, Figures 14-16 are unnecessary and do not bring any new knowledge.
As it was shown by Authors, the front-end band-limiting of the signal causes a frequency dependant bias in the measurement that needs to be calibrated for and removed. This is the only shortcoming of the described approach, but it can be accepted at this stage of research in my opinion.
Author Response
Dear Sir/Madam,
My I first say a very big thank you for taking the time to read and review our article. I understand that your time is limited and valuable and am very grateful that you have chosen to spend yours assisting us on this journey. I will address the points you have raised in bullet point form below. All changes to the manuscript have been highlighted in bold for your convenience.
- “in irrelation” has been amended to “in relation”
- A reference which examined the DET and its 2-D correlation function has been included. In this paper it was shown that there exists a small skew in the correlation function which indicates the two loops are not truly independent. However, this skew is small and thus so is the covariance. The full equation for the measurement variance includes a -2cov(X,Y) term and thus neglecting this results in an overestimate which we feel is satisfactory for threshold calculation.
- Figures 14-16 have been integrated into a single file at the behest of another reviewer and we feel demonstrate the threshold selection process visually very well.
- Thank you for your kind words.
Round 2
Reviewer 1 Report
Comments and Suggestions for Authors
The concerns have been addressed, I think the MS in current form is acceptable.